

# Global vegetation variability and its response to elevated $CO_2$, global warming, and climate variability - A study using the offline SSiB4/TRIFFID model and satellite data

Ye Liu[1], Yongkang Xue[1*], Glen MacDonald[1], Peter Cox[2], Zhengqiu Zhang[3]

[1] University of California Los Angeles (UCLA), Los Angeles, CA, USA
[2] College of Engineering, Mathematics and Physical Science, University of Exeter, Exeter, UK
[3] Chinese Academy of Meteorological Sciences, Beijing, China

*Correspondence to*: Yongkang Xue (yxue@geog.ucla.edu)

**Abstract.** The climate regime shift during the 1980s had a substantial impact on the terrestrial ecosystems and vegetation at different scales. However, the mechanisms driving vegetation changes, before and after the shift, remain unclear. In this study, we used a biophysical-dynamic vegetation model to estimate large-scale trends in terms of carbon fixation, vegetation growth and expansion during the period 1958-2007, and to attribute these changes to environmental drivers including elevated atmospheric $CO_2$ concentration (hereafter $eCO_2$), global warming, and climate variability (hereafter CV). Simulated Leaf Area Index (LAI) and Gross Primary Product (GPP) were evaluated against observation-based data. Significant spatial correlations are found (correlations>0.87), along with regionally varying temporal correlations of 0.34-0.80 for LAI and 0.45-0.83 for GPP.

More than 40% of the global land area shows significant trends in LAI and GPP since the 1950s: 11.7% and 19.3% of land has consistently positive LAI and GPP trends, respectively; while 17.1% and 20.1% of land, saw LAI and GPP trends respectively, reverse during the 1980s. Vegetation fraction cover (FRAC) trends, representing vegetation expansion/shrinking, are found at the edges of semi-arid areas and polar areas.

Overall, $eCO_2$ consistently contributes to positive LAI and GPP trends in the tropics. Global warming is shown to mostly affected LAI, with positive effects in high latitudes and negative effects in subtropical semi-arid areas. CV is found to dominate the variability of FRAC, LAI, and GPP in the semi-humid and semi-arid areas. The $eCO_2$ and global warming effects increased after the 1980s, while the CV effect reversed during the 1980s. In addition, plant competition is shown to have played an important role in determining which driver dominated the regional trends. This paper presents a new insight into ecosystem variability and changes in the varying climate since the 1950s.

**Keywords**: Ecosystem variability, dynamic vegetation modeling, elevated CO2, global warming, climate change and variability, TRIFFID, SSiB





## 1 Introduction

Climate variability and change, including global warming, and elevated atmospheric $CO_2$ concentrations (referred to as $eCO_2$ in this paper), have profound impacts on the terrestrial biosphere at global and regional scales (e.g. Garcia et al., 2014); while the terrestrial biosphere, in turn, affects the global climate by altering fluxes exchanges, energy balance, carbon cycle,

etc. (e.g. Cox et al., 2000;Xue et al., 2004;Xue et al., 2010;Ma et al., 2013;Friedlingstein et al., 2006). Important trends in terrestrial ecosystem carbon fixation, growth, and expansion in the past 60 years have been detected. For instance, general earth greening has been discovered by analyzing satellite-derived Normalized Difference Vegetation Index (NDVI) (Myneni et al., 1997;Ichii et al., 2013;Los, 2013;Piao et al., 2011) and Leaf Area Index (LAI, defined as the one-side leaf area in a unit area) products (Piao et al., 2011;Piao et al., 2015;Zhu et al., 2016). The Earth's terrestrial vegetation has acted as an

important carbon sink in the past 60 years (Ballantyne et al., 2012;Le Quéré et al., 2013), with a significantly increasing rate after the 1980s (Sitch et al., 2015), revealing growth in plant productivity (Nemani et al., 2003;Anav et al., 2015). In the meantime, vegetation fractional coverage (hereafter FRAC) has been changing, including some large-scale increases in total vegetation cover (e.g. Piao et al., 2005;Donohue et al., 2009;McDowell et al., 2015), and shifts in the spatial distributions of plants species, such as woody plants encroachment in the savanna area (Stevens et al., 2017) and shrubification in the tundra

biome (Epstein et al., 2012;Mod and Luoto, 2016).

Many studies have attributed these large-scale ecosystem trends to climatic drivers and $eCO_2$ after applying statistical methods to satellite-based observations or the results from process-based land surface models (e.g. Myneni et al., 1997;Liu et al., 2006;Ichii et al., 2013;Piao et al., 2015;Schimel et al., 2015;Sitch et al., 2015;Devaraju et al., 2016;Zhu et al., 2016;Smith et al., 2016;Mao et al., 2013). Statistical regression and cross-correlation have been applied to attribute the

recent biosphere changes to precipitation, temperature, and solar radiation variability (e.g. Zeng et al., 2013;Myers-Smith et al., 2015). Results from these studies indicated that northern mid- to high- latitude NDVI anomalies were positively correlated with temperature, and positively associated with precipitation in temperate to tropical semi-arid and arid regions (Zeng et al., 2013). However, statistical methods rarely isolate the drivers' contribution to the inter-annual or decadal variability of the terrestrial ecosystem (e.g. Ahlbeck, 2002;Piao et al., 2015). Moreover, the satellite products only cover the

period after 1980.

Process-based land surface models overcome these limitations and are also able to include $CO_2$ as an external driver. Dynamic Global Vegetation Models (DGVMs) predict vegetation cover changes in response to changes in climate and $CO_2$, and update associated surface characteristics (Smith et al., 2001;Bonan et al., 2002;Sitch et al., 2003;Woodward and Lomas, 2004;Krinner et al., 2005;Zeng et al., 2005;Zaehle and Friend, 2010;Lawrence et al., 2011;Zhang et al.,

2015;Claussen and Gayler, 1997). By applying DGVMs, a general consensus has been reached that $eCO_2$ explains the greater part of the increasing trend of LAI and GPP since the later 1980s. Air temperature, precipitation, land use and land cover change, and nitrogen decomposition, also play roles in the changing terrestrial biosphere (e.g. Cramer et al., 2001;Schimel et al., 2015;Zhu et al., 2016). However, DGVMs should be applied with caution. The Coupled Model





Intercomparison Project Phase 5 (CMIP5) reported that most DGVMs overestimated LAI in comparison to Global Inventory Monitoring and Modeling System (GIMMS) data (Murray-Tortarolo et al., 2013;Zhu et al., 2013). In addition, large discrepancies between models were found when predicting ecosystem variability and trends (Piao et al., 2013;Zhu et al., 2017). Unsurprisingly, the dominant factors obtained from different models are often significantly different (Beer et al.,

2010;Huntzinger et al., 2017). Furthermore, DGVM simulations were sensitive to meteorological forcing data (Slevin et al., 2017;Wu et al., 2017). Therefore, a comprehensive evaluation of large-scale terrestrial ecosystem vegetation trends and potential drivers is crucial for improved DGVM application.

Most ecosystem trend detection and attribution studies have focused on the period after the 1980s when satellite data has been available. However, a climate regime shift, identified by abrupt shifts in temperature, precipitation, and other

external forcing, was observed during the 1980s (e.g. Gong and Ho, 2002;Lo and Hsu, 2010;Reid et al., 2016). The responses of vegetation to these climate shifts have not yet been comprehensively investigated, especially at the level of individual Plant Function Types (PFTs).

In this study, we investigate the effect of $eCO_2$ and climate drivers including global warming and climate variability (meteorological forcing excluding global warming, referred to as "CV") on the trends of FRAC, LAI, and GPP during the

period 1958-2007 by applying a dynamic global vegetation model (Simplified Simple Biosphere model version 4/Top-down Representation of Interactive Foliage and Flora Including Dynamics, SSiB4/TRIFFID, Xue et al., 1991; Cox, 2001; Zhan et al., 2003; Zhang et al., 2015; Harper et al., 2016) at both grid and PFT levels, and using satellite products whenever they are available. Changes in the ecosystem trends are attributed to changes in $eCO_2$ and climate effects, focusing particularly on the climate regime shift during the 1980s. The model and data description are presented in section 2. In section 3, we show

results of a quasi-equilibrium simulation, which is used to produce the initial condition for the subsequent long-term simulation. In section 4, the model performance in reproducing the spatial distribution and temporal evolution of vegetation variables are evaluated using satellite products. Finally, the paper delineates the linear trend of FRAC, LAI, and GPP before/after the 1980s and quantifies the contribution of different environmental drivers.

## 2. Model description, observational datasets, and experimental design

### 2.1 Brief description of SSiB4/TRIFFID

The Simplified Simple Biosphere model (SSiB) is a biophysically based model incorporating estimates fluxes of radiation, momentum, sensible heat, and latent heat, as well as runoff, soil moisture, and surface temperature (Xue et al., 1991). A photosynthesis model (Collatz et al., 1991;Collatz et al., 1992) has been implemented into SSiB to calculate carbon assimilation, forming SSiB2 (Zhan et al., 2003). The Top-down Representation of Interactive Foliage and Flora Including

Dynamics (TRIFFID) (Cox, 2001;Harper et al., 2016) was subsequently coupled with SSiB version 4 to calculate vegetation dynamics, including relevant land-surface characteristics of vegetation cover and structure (Xue et al., 2006), called SSiB4/TRIFFID. Zhang et al. (2015) updated the competition dominance hierarchy from tree-shrub-grass (i.e., trees



dominate shrubs and grasses, and shrubs dominate grasses) to tree-grass-shrub, but still allowed shrubs and grasses to compete for sunshine and space. Some parameters were also updated in this process. In a North American study (Zhang et al., 2015) the SSiB4/TRIFFID reproduced the main ecosystem features of North America. Based on the North America experiment, we further distinguished deciduous/evergreen broadleaf trees as they form the land-cover in different latitudes and have distinctly different phenological features. The absence of deciduous broadleaf trees in SSiB4/TRIFFID caused an

unrealistic lower summer LAI in the northeastern U.S. (Zhang et al., 2015). In addition, tundra was separated from the original single shrub category in order to better reflect the arctic biomes. To date, SSiB4/TRIFFID therefore includes 7 PFTs: 1) Evergreen broadleaf trees, 2) Deciduous broadleaf trees, 3) Needle leaf broadleaf trees, 4) C3 grasses, 5) C4 plants, 6) Shrubs, and 7) Tundra.

**2.2 Meteorological forcing data**

The Princeton global meteorological dataset for land surface modeling (Sheffield et al., 2006) was used to drive SSiB4/TRIFFID for the period of 1948-2007 at $1^{o}$ x $1^{o}$ spatial resolution and 3-hourly temporal interval. This dataset, including surface air temperature, pressure, specific humidity, wind speed, downward short-wave radiation flux, downward long-wave radiation flux, and precipitation, is constructed by combining a suite of global observation-based datasets with the

National Centers for Environmental Prediction/National Center for Atmospheric Research reanalysis data.

**2.3 Observation-based data**

The global land cover map products from the Global Land Cover (GLC) database for the year 2000 is produced by an international partnership of about 30 research groups coordinated by the European Commission's Joint Research Centre (Bartholome et al., 2002). It is based on daily S1 data from the VEGETATION sensor on-board SPROT 4 acquired between

$1^{st}$ November 1999 and $31^{st}$ December 2000. This dataset provides a global map with one consistent legend, as well as regional maps with separate legends containing more detail for certain regions. For instance, tundra is not included in the 21-category global legend but is included in the regional product for Northern Eurasia (Bartalev et al., 2003). In this study, we generated a GLC2000 dominance map for land cover validation according to the regional maps and the corresponding legends. Other than GLC2000, we also employed the Moderate Resolution Imaging Spectroradiometer (MODIS) land cover

data product (MCD12Q1) as a reference (Friedl et al., 2010).

To access the climatology, variation, and trends of simulated LAI, two widely used LAI products were used as references in this study: the Global Inventory Modeling and Mapping Studies (GIMMS) LAI and the Global LAnd Surface Satellite (GLASS) LAI. GIMMS-LAI is based on the third generation of Normalized Difference Vegetation Index (NDVI3g) from the GIMMS group and an Artificial Neural Network model (Zhu et al., 2013). GIMMS-LAI provides a 1/12-degree

resolution, 15-day composites, and spans the period July 1981 to December 2011. GLASS-LAI is generated from Advanced Very High Resolution Radiometer (AVHRR) (from 1982 to 1999 with 0.05-degree resolution) and MODIS (from 2000 to 2012 with 1 km resolution) time-series reflectance data using general regression neural networks (Xiao et al., 2014). GIMMS



and GLASS LAI, and the meteorological forcing data for overlap period 1982 to 2007, were remapped to 1-degree spatial resolution and a monthly temporal interval.

The Model Tree Ensemble (MTE) GPP product (Jung et al., 2009;Jung et al., 2011) was used as the reference to evaluate the GPP simulation. MTE is based-on a machine learning technique in which the model is trained to predict the

carbon fluxes at FLUXNET sites driven by observed meteorological data, land cover data, and the remotely-sensed fraction of absorbed photosynthetic active radiation (Jung et al., 2009). Then the trained model is then applied at grid scale driven by gridded forcing data. MTE-GPP data was remapped to 1-degree spatial and a monthly temporal resolution.

### 2.4 Experimental design

In this study, SSiB4/TRIFFID was applied to produce the global vegetation distribution and assess the sensitivity of

ecosystem trends to climate and $eCO_2$. For this purpose, two sets of simulations were conducted: 1) a 100-year quasi-equilibrium simulation driven by climatological forcing, and 2) sensitivity simulations driven by real-forcing from 1948-2007 (**Figure 1**). In the first set, SSiB4/TRIFFID was driven with the climatological forcing and 1948 $CO_2$ concentration to reach a steady state, which was used as the initial condition in the second set of simulations. Meanwhile, the effect of large-scale disturbance (LSD) on restricting tree expansion to savanna areas was investigated.

Using the quasi-equilibrium simulation results as the initial condition, the historical meteorological forcing and yearly updated $CO_2$ concentration were used to drive SSiB4/TRIFFID from 1948 through 2007. In this control simulation, we firstly evaluated the model performance in reproducing the climatology and variability of multiple biotic variables in comparison with multiple observation-based datasets. The long-term trends were diagnosed before and after the climate regime shift of the 1980s. Furthermore, three sets of experiments were conducted to quantify the effects of external drivers

(climate and $CO_2$) and vegetation competition on the ecosystem trends. These experiments were designed as following:

1. Fixed-$CO_2$: The model was driven by the same meteorological forcing as the control experiment, but $CO_2$ concentration was fixed at the level of 1948 (310.33 ppm). The difference between control experiment and Fixed-$CO_2$ indicates the $eCO_2$ effect.

2. Detrend-Temp: The mean warming trend over each 10 degrees of latitude, from 1948 to 2007, was subtracted in

this experiment. Then the detrended temperature along with other meteorological forcing and annually varying $CO_2$ concentration were used to drive the model. Subtraction of Detrend-Temp from the control experiment isolates the effect of global warming.

3. Climate Variability: Subtraction of both Fixed-$CO_2$ and Detrend-Temp from the control experiment was regarded as representing the effect of CV.



### 3. Vegetation initial conditions

#### 3.1 Quasi-equilibrium simulation

DGVMs typically requires a number of state variables as an initial condition pertaining to a particular meteorological forcing. DGVMs commonly takes 50-1000 years' simulation to reach this steady-state (e.g. Bonan and Levis, 2006;Zeng et al., 2008). Since our purpose was only to generate the initial condition for the decadal simulations, we applied a shortcut to reach the quasi-equilibrium coexistence of PFTs under the climatological forcing. We started the model from a SSiB 1-degree dominant vegetation map (Xue et al., 2004), with 100% occupation of the dominant PFT and zero for other PFTs. The 1948-2007 averaged meteorological forcing along with 1948 $CO_2$ concentration was used to drive the SSiB4/TRIFFID for 100 years.

SSiB4/TRIFFID is a water, energy, and carbon balanced model. Plants expansion, and biotic properties such as vegetation height and LAI are constrained by carbon reallocation. FRAC links the carbon accumulation within plants and intra-species competition via a system of Lotka-Volterra equations (Cox, 2001). **Figure 2** shows the temporal evolution of FRAC during the quasi-equilibrium simulation over four general climate zones. In the early first ten years of the simulation, most PFTs' FRACs change rapidly, but then take decades to reach a steady state. To qualify the steady state, we define quasi-equilibrium status as occurring when the rate of change of vegetation fraction is less than 2% of the mean vegetation fraction, over the last ten years of simulation. The result shows 100-year spin-up time is sufficient for our model.

Overall, in the tropical areas (23.5°S~23.5°N), C4 plants and evergreen broadleaf trees are of mixed dominance and coexistent with C3 grasses, shrubs, and deciduous broadleaf trees. The subtropical areas (23.5°~35° in both hemispheres) are dominated by C3 grasses, C4 plants, and shrubs with similar occupation for each (~18%), whereas 40% of the subtropical areas are occupied by bare land. Needle leaf trees, C3 grasses, shrubs, and deciduous broadleaf trees are mix dominant the temperate zones (35°~66.5°, particularly in North Hemispheres). Over the polar areas (66.5°~90° in both hemispheres), shrubs and tundra are of mixed dominance (**Figure 2**). We recognize that these broad zones each incorporate many different PFTs that are not represented in our model. That being said, the broad distributions are very close to those based on the climatology of the global ecosystem simulated by the SSiB4/TRIFFID using the meteorological forcing from 1958-2007, which will be discussed in detail in Section 4.1.1

#### 3.2 Effect of large-scale disturbance

Our preliminary quasi-equilibrium run showed that the absence of fire disturbance in SSiB4/TRIFFID causes trees' extension into the South American and African savanna areas (**Figure 3a**), where the climate acting alone would seem to favors tree growth (Bond et al., 2005). However, disturbance alters the physical structure or arrangement of biotic elements quickly and with great effect. Major ecological disturbances such as fire, flooding, and insect outbreaks vary spatially and temporally. TRIFFID prescribes the large-scale disturbance (LSD) coefficient as a uniform value for every PFT globally according to model calibration (Cox, 2001). We raised the LSD coefficient (largely representing fire at this scale) from 0.004





to 0.04 for tree PFTs that coexist with C3 grasses and C4 plants. With the updated setting, SSiB4/TRIFFID produced reasonable dominant tree coverage over the tropical rainforest areas (**Figure 3b**) in comparison with the GLC2000 dataset. The global PFT distributions in the equilibrium run are close to the results using the real meteorological forcing. Detailed comparison of the simulated global vegetation spatial distribution with the contemporary satellite-based datasets is discussed
in Section 4, in which the model was driven by the 1948-2007 meteorological forcing.

### 4. Simulation results using the 1948-2007 meteorological forcing

Based on the equilibrium run as initial condition for both biotic and abiotic variables such as FRAC, LAI, vegetation height, soil moisture, and temperature, the model was then driven with the historical meteorological forcing and yearly updating $CO_2$ concentration from 1948 to 2007. DGVMs have shown diverse performance in reproducing the spatial distribution and
temporal variability of the ecosystems (Murray-Tortarolo et al., 2013;Piao et al., 2013;Anav et al., 2015;Zhu et al., 2017), which resulted in large discrepancies between models in identifying the attributed dominant drivers of changes (Beer et al., 2010;Huntzinger et al., 2017). It is therefore important to validate the model performance first before use for attribution studies. In this section, the global (hereafter referring to the regions of 180°W to 180°E, 60°S to 75°N) distributions of the simulated FRAC, LAI, and GPP are compared to the observation-based datasets. LAI and GPP inter-annual variabilities are
also evaluated over several regions, showing large ecosystem variability and changes.

### 4.1 Assessment of the simulated vegetation spatial distribution

It is a challenge for DGVMs to reproduce PFT coexistence, particularly for the smaller PFTs in the semi-humid to semi-arid areas as they are fragile and sensitive to climate and vulnerable to competition (Fu et al., 2006). For instance, Zeng et al. (2008) introduced a specific sub-model to grow the trees and grasses firstly, then shrubs in unoccupied spaces. However,
recent studies suggest that the interannual variability of the global terrestrial carbon sink is dominated by semi-arid ecosystems (Ahlstrom et al., 2015). The SSiB4/TRIFFID allows smaller stature PFTs to co-exist (Cox, 2001;Zhang et al., 2015).

### 4.1.1 Global Vegetation distribution

The spatial distribution of the dominant PFT is closely related to large-scale the climate (MacDonald, 2002). **Figure 4** shows
the FRAC for each PFT from the control experiment averaged over the last ten years of the run (1998-2007). Overall, the model simulated vegetated land cover of 77.7% of the global land surface, close to the GLC2000 estimation, 76.0%, and higher than the MODIS estimation, 73.8%. There was no human activity included in the model simulation, as such an agricultural category was not included. Therefore, in the following vegetation coverage comparison with the GLC products, PFT coverage in the gridboxes with agriculture was reduced by multiplying the agriculture fraction. On this basis, the total
simulated tree cover (the sum of evergreen broadleaf trees, deciduous broadleaf trees, and needle leaf trees) was 25.0%,



close to 24.6% in GLC2000. The evergreen broadleaf trees in the Amazon, Central Africa, and Southeast Asia, deciduous broadleaf trees in southeast North America, and needle leaf trees in the high-mid latitudes of North America and Eurasia were reasonably predicted (**Figure 4-5**). SSiB4/TRIFFID predicted 12.1% C3 grass occupation, which is slightly lower than 12.5% in the GLC2000, with reasonably simulation in the mid-latitudes in both hemispheres such as the central U.S.,

Eurasian Steppes, South America, South and East Africa, and East Australia (**Figure 5**). The model predicted 9.8% natural C4 plants, compared to 8.3% in the GLC2000. The discrepancy could be partially attributed to the absence of C4 plants in some GLC2000 regional maps (such as Southeast Asia).  The global GLC2000 map is assembled from these regional maps. In fact, a satellite-based physiological model simulation predicted 13.9% of C4 plants coverage with no agriculture category (Still et al., 2003). In the SSiB4/TRIFFID prediction without excluding agricultural land, the C4 plants cover 13.2%, close to

Still et al's estimation. C4 plants are primarily located in South American and African savanna areas, the Indian Subcontinent, Southeast Asia, the southeastern U.S., and northern Australia. The model predicted 15.6% shrubs and tundra occupation, which was close to 15.4% in the GLC2000, with shrubs primarily located in the semi-arid areas in both hemispheres and the pan-arctic area, while tundra was located in the pan-arctic area and Tibetan Plateau.

**4.1.2 Leaf area index and Gross Primary productivity**

This section discusses the spatial and temporal correlations between the simulations and observations and compares with other model results.  Since other published studies on this subject have not excluded agricultural areas, to make our results comparable with others, the agricultural areas were not subtracted.  Moreover, the difference between the results presented in this section and those excluding the agriculture area are less than 0.01.

SSiB4/TRIFFID produced a similar global LAI pattern compared to both the GIMMS and GLASS products,
confirmed by global spatial correlation coefficients of 0.86 (GIMMS, $p<0.05$) and 0.87 (GLASS, $p<0.05$), and above 0.74 ($P<0.05$) against both observations over the Northern Hemisphere (**Figure 6**). Previous studies reported spatial correlation coefficients between models and GIMSS-LAI over the globe/Northern Hemisphere in the range of 0.44-0.77/0.21-0.61 (Murray-Tortarolo et al., 2013;Mahowald et al., 2015). The latter study reported that in general DGVMs tended to overestimate global average LAI by 0.69±0.44 units [recalculated from (Mahowald et al., 2015)]. SSiB4/TRIFFID produced
~0.95 units higher global averaged LAI than the satellite-based data. The absence of nitrogen limitation in the model could contribute to the overestimation and the latest development of SSiB5/TRIFFID with a global nitrogen model seems promising.

The spatial correlation coefficient between our simulation of GPP and the MTE dataset is 0.93 ($P<0.05$) (**Figure 7**). Anav et al. (2015) reported less than 0.8 correlation against MTE-GPP for multi-model comparison. Over the globe,
SSiB4/TRIFFID predicted 1135 $gC/m^2/yr$ (or 151 PgC/yr), greater than the MTE average of 920 $gC/m^2/yr$ (or 122 PgC/yr). However, our simulation was in the range of 130-169 PgC/yr reported by Anav et al. (2015) and 111-151 PgC/yr reported by Piao et al. (2013) for 10 offline models. In addition, to our model's deficiencies (such as lack on N-limitation), the lack of $CO_2$ fertilization during MTE model training may have contributed to an underestimation in MTE-GPP.



### 4.2 Assessment of the simulated vegetation temporal variability

Model performance in predicting temporal variability has been less evaluated in previous studies on ecosystem trend detection and attribution (e.g. Ichii et al., 2013;Piao et al., 2013;Zhang et al., 2015). However, performance in predicting LAI and GPP trends and variability has been found to vary among models (Murray-Tortarolo et al., 2013;Piao et al., 2013;Anav

et al., 2015;Zhu et al., 2017). To better assess model performance in this regard, we selected 13 sub-regions associated with different regional climate and land cover conditions (**Table 1**). Although the MTE excludes $CO_2$ fertilization during its model training, MTE-GPP still incorporates variability at different scales associated with climate variability and is widely used by the community for the model validation. Both annual LAI and GPP correlation coefficients were calculated over the period of 1982-2007.

Globally, the correlations for annual mean LAI between the SSiB4/TRIFFID and the satellite-based products were 0.58 (P<0.05) for GIMMS and 0.64 (P<0.05) for GLASS. The correlation for annual GPP was 0.59 (P<0.05) between the SSiB4/TRIFFID and MTE GPP. Piao et al. (2013) had reported less than 0.4 correlation by for GPP from 10 offline-models. This improvement may be due to SSiB4/TRIFFID better capturing the interannual variability in semi-arid areas which dominate interannual variability (Ahlstrom et al., 2015). For instance, LAI correlations over West Africa were 0.79 (P<0.05)

with GIMMS and 0.77 (P<0.05) with GLASS **(Figure 8)**, and GPP correlation was 0.80 (P<0.05) with MTE in that region. For the other semi-arid areas in western North America, South American savanna areas, and East Africa, the simulated LAI significantly matches at least one of the two reference datasets with correlation in the range of 0.46-0.58 (P<0.05). The simulated GPP correlations with MTE were in the range of 0.63-0.70 (P<0.05).

      For the forested areas, simulated LAI and GPP were both well correlated with reference data over the Northern

Hemisphere boreal forests, whereas only LAI had significant correlation over the rainforest areas. The inconsistency over the rainforests indicated that the missing $CO_2$ fertilization in MTE-GPP could be a predominant limitation to GPP in those areas. Over the cold regions (subarctic and Tibetan Plateau), SSiB4/TRIFFID matched the reference data closely with the significant correlations ranging 0.35-0.74 (P<0.1) for LAI and 0.49-0.72 (P<0.05) for GPP. It should be pointed out that although there is general consistency between the satellite-based LAI products, large relative uncertainties were identified

(Jiang et al., 2017).

      As shown in **Figure 8**, there were distinct decadal variabilities in most sub-regions. For instance, trends reversal sign in West Africa (from negative to positive) and western North America (from positive to negative). Areas with enhancement in trends slopes, such as the subarctic after the 1980s climate regime shift, will be discussed in detail in the next section.

Summary, statistics are listed in **Table 2**. Generally, SSiB4/TRIFFID presented reasonable predictions of terrestrial ecosystem climatology and variability compared to the observation-based datasets. Compared to other DGVMs, SSiB4/TRIFFID showed above average performance in reproducing the spatial distribution, but with certain bias in absolute





numbers. In particular, SSiB4/TRIFFID captured the ecosystem variabilities over different regions across the world, which provides a good basis for pursuing the ecosystem trends detection and attribution study presented in the next section.

**5. Detection and attribution of decadal trend change during the 1980s**

There are a number of studies showing abrupt environmental change during the 1980s (e.g. Hare and Mantua, 2000;Lo and

Hsu, 2010;Reid et al., 2016). Since the 1950s, global land surface temperature has risen with the rate of increase accelerating after the 1980s (**Figure 9**). The mean land surface temperature during the three decades of the 1950s, 1980s, and 2000s was 286.84 K, 286.97 K, and 287.65 K, respectively. Global precipitation had shown distinct decadal variability.  It decreased from 2.34 mm/day in the 1950s to 2.28 mm/day in the 1980s, then recovered to 2.33 mm/day in the 2000s. In the meantime, the atmospheric $CO_2$ concentration steadily increased from 313.44 ppm in the 1950s to 373.67 ppm in the 2000s. In this

section, we will discuss how the changing climate and $eCO_2$ affected the large-scale vegetation. The sensitivity experiments described in section 2.1 were conducted to quantify the contributions of $eCO_2$, global warming, and CV on the trends of FRAC, LAI, and GPP at global and regional scales and on PFT variation and competition.  The first ten-years of each simulation (1948-1957) were excluded from the analysis as this served as the spin up period.

**5.1 Three major types of trend change since the 1950s**

The climate regime including temperature and precipitation shifted abruptly during the 1980s, inevitably giving rise to changing ecosystem trends in many parts of the world. Here we compare trends of FRAC, LAI, and GPP over two periods: 1958-1982 and 1982-2007. The model performance for the second period, for which satellite observations are available, was evaluated in Section 4. Spatial patterns of the trends are shown in **Figure 10**, in which the dots indicate the grid points with a p-value less than 0.1 according to a Mann-Kendal test (Mann, 1945;Kendall, 1955).

20        At the global scale, significant vegetation trends were only found in the simulations after the 1980s. During this period, FRAC increased at the rate of 0.032/yr. LAI had a positive trend of 0.0029/yr, which matched very well to GIMMS' results (0.0029/yr). GPP had a positive trend of 2.22 $gC/m^2/yr^2$, within the range of 1.60-4.69 $gC/m^2/yr^2$ for GPP over similar periods (e.g. Anav et al., 2015;Yue et al., 2015).  In contrast to LAI and GPP, there were relatively few areas with a significant simulated FRAC trend.

25        For the global land surface, over 40.2% had a significant LAI trend since 1958, and over 48.1% had a significant GPP trend. In response to the climate regime shift during the 1980s, the terrestrial ecosystem had three major trend changes in different parts of the world after the 1980s (**Table 3**). 1) There was trend sign reversal from negative to positive in the East Asian monsoon area, West Africa, Central Asia, and Eastern US, over 14.2% (LAI) and 11.4% (GPP) of the land surface. In particular, West Africa experienced the largest vegetation deterioration in the world before the 1980s, associated

with LAI and GPP reductions of 0.0258/yr and 18.54 $gC/m^2/yr^2$, respectively - approximately 10 times the trends of the global average. After the 1980s, recovery is simulated at the rate of 0.0137/yr and 8.02 $gC/m^2/yr^2$ for LAI and GPP,



respectively. 2) Trend sign reversal from positive to negative was found in western North America, South America savanna and East Africa, which accounted for 2.9% (LAI) and 2.7% (GPP) of the land surface. 3) There were consistent positive trends substantially enhanced by at least 50% of prior period trends after the 1980s, which were found in Equatorial rainforest areas, boreal forest areas, South Africa, North Australia, subarctic areas, and the Tibetan Plateau, representing over

11.7% (LAI) and 19.3% (GPP) of the land surface. There were also areas with consistent positive trends but no substantial change during the entire period or other types of trend change over much smaller areas. The first three major trend changes as described above will be discussed in the following sections.

### 5.2 Global overview of three simulated external forcing effects on the ecosystem trends

The differences between the control experiment and Fixed-$CO_2$ shows that e$CO_2$ stimulated vegetation growth mainly in the

Equatorial areas and eastern North America, Western Europe, and Eastern China in the mid-latitudes. Substantially enhanced positive trends were found after the 1980s for both LAI and GPP over those areas (**Figure 11**). e$CO_2$ promoted rainforest LAI increase only after the 1980s; however, its effect on GPP appeared during the entire period. GPP was directly linked to $CO_2$ through the photosynthesis process, while LAI, in addition to the photosynthesis process, was also affected by respiration and carbon reallocation in plants, which were influenced by climate and e$CO_2$ (O'Sullivan O et al., 2017).

15          The differences between the control experiment and Detrend-Temp shows that global warming had minor effects on the trends of LAI and GPP before the 1980s (**Figure 12**). After the 1980s, the rapidly enhanced warming contributed positive LAI trends at high latitudes, while the GPP change seems less substantial. Meanwhile, there were negative trends due to heat stress in low latitudes, particularly in the semi-arid regions such as South American savanna, East Africa, and central Asia.

20          The differences between the control experiment and Fixed-$CO_2$ and Detrend-Temp show the CV effect, which has complex influences on the ecosystem. The CV in this study includes contribution of changes in surface pressure, precipitation, surface wind speed, downward longwave and shortwave radiations, surface humidity, along with temperature that exclude the trend. Precipitation was however found to play a dominant role. The correlation coefficients between the annual mean CV effect on LAI and GPP and annual mean precipitation at the grid points with significant CV effect were

greater than 0.60 (P<0.05). Overall, the CV effect alone can explain the total FRAC trends in the control experiment (**Figure 13**).

          Before the 1980s, CV caused LAI decrease in East Asian monsoon areas, eastern North America, West Africa, Western Europe, Central Asia, Siberia, and eastern Australia. The GPP also decreased in these areas except for eastern North America, Western Europe, and Siberia. In contrast, the CV effect before the 1980s led LAI and GPP increase in the Tibetan

Plateau and South Asia, western North America, South American savanna areas, East and South Africa, and northern Australia. Due to the climate regime shift, CV had produced the opposite sign to the trends of LAI and GPP in East Asian monsoon areas, Central Asia, West Africa, North America, South American savanna areas, and East Africa. In some areas,



such as South Africa and northern Australia, persistent precipitation increase/decrease led to sustained positive/negative trends from the 1950s.

Overall, after the 1980s, the effects of eCO$_2$ and global warming have been generally enhanced; but the CV effect has exhibited distinctly different regional features before/after the 1980s over many regions in the world.

## 5.3 Assessments of synthesized effects of the three external forcings in influencing the regional ecosystem trends

The discussion in this section is based on **Tables 3** and **4** and **Figure 14**. Only the significant changes listed in the tables are discussed and the following text is restricted to a subset of world regions – West Africa and East Asia; western North America, and rainforest, boreal forest, subarctic and Tibetan Plateau.

### 5.3.1 Dominant factor in influencing trend reversal from negative to positive in West Africa and East Asia

CV was found to be the dominant driver of the ecosystem trends in West Africa, explaining most of the LAI and GPP trends and trend changes. Before the 1980s, CV caused C4 plants' LAI, GPP, and FRAC over the region to decrease, followed by shrubs, whereas eCO$_2$ caused C3 grasses' LAI and GPP to slightly increase. Global warming showed little effect during the entire period from 1958-2007. The ecosystem trends in West Africa reversed to increase when the precipitation trend changed to increase after the 1980s, with the major increase in C3 grasses and shrubs over the region. A previous study using satellite data also showed recent West Africa greening was highly correlated to the precipitation increase (e.g. Herrmann et al., 2005). The response to eCO$_2$ of a particular PFT not only depends on its own physiological and morphological characteristics, but is also determined by the interactions that arise with the other PFTs, which are competing for the same resources. eCO$_2$ played a role in increasing C3 grasses coverage since the 1950s. However, the PFT competition outcomes reduced the C4 plant coverage over the region, mainly after the 1980s when eCO$_2$ had a large impact. As such, the change in regional FRAC overall within West Africa is not significant and has been compromised by positive and negative contributions of the individual PFTs after the 1980s. However, our results show that the boundary between Sahara and Sahel has experienced significant variation since the 1950s. Based on the observed precipitation data and the precipitation/NDVI correlation, Thomas and Nigam (2018) suggest a Sahara Desert expansion since the 1950s. Our results are in an agreement at large with the Thomas and Nigam's study (2018) but also with substantial differences. A comprehensive discussion on this issue is out of scope of this paper and will be addressed in a separate paper.

Regional average trends reversed in the East Asian monsoon area because CV and eCO$_2$ dominated LAI and GPP trends, before and after the 1980s, respectively. Their combined effects caused trend reversal in the East Asian monsoon areas. The eCO$_2$ dominated the PFT LAI and GPP trends since the 1950s, which caused significant increase in C3 grasses and trees but significant decrease in C4 plants (**Figure 14**). Field experiments reported that the differential growth and competitiveness responses of C3 and C4 plants to eCO$_2$ is complex and under debate (Lee, 2011;Miri et al., 2012;Leakey et al., 2009). Our simulations suggest that there is a potential increase C3 grasses long-term competitive ability at regional scale. Meanwhile, enhanced global warming after the 1980s stimulated C4 plant growth, but this effect was compromised by



its detrimental effect on C3 grasses after the 1980s. Furthermore, CV contributed decreasing trends of LAI and GPP before the 1980s, but with minor effects after the 1980s. Overall, CV and $eCO_2$ dominated the negative and positive trends in this area before and after the 1980s, respectively.

**5.3.2 Dominant factor in influencing trend reversal from positive to negative in western North America**

The $eCO_2$ effect persistently caused LAI, GPP, and FRAC increase since the 1950s, while global warming reduced both LAI and GPP only after the 1980s. However, CV dominated the LAI and GPP trends and trends reversal in western North America by causing the dominant PFTs (C3 grasses and shrubs) to increase/decrease before/after the 1980s (**Figure 14** and **Table 4**). The CV effect on FRAC change was more complex due to its different effects on LAI and GPP and FRAC expansion in C3 and shrub PFTs after the 1980s (**Figure 14**): both C3 and shrubs expanded with LAI and GPP decrease.

This discrepancy suggests that expansion might be coupled with carbon fixation less than with growth in the model. We conjecture that CV may promote vegetation expansion into some areas that were largely unvegetated, but this requires further investigation.

**5.3.3 Dominant factor in influencing the enhanced positive trend in rainforest, boreal forest, subarctic, and Tibetan Plateau**

$eCO_2$ and CV had persistent positive impacts on rainforest growth in terms of LAI and GPP since the 1950s (**Figure 14**). $eCO_2$ dominated the LAI and GPP trends in both periods except for the LAI positive trend before the 1980s, which was dominated by CV. LAI trend enhancement after the 1980s was associated with increased $CO_2$ fertilization, while GPP trend enhancement was attributed to increase in both $eCO_2$ and CV effects (**Table 4**). The importance of $CO_2$ and CV impacts on the rainforests was confirmed by previous analyses on the trends of LAI and NDVI (e.g. Hilker et al., 2014;Zhu et al., 2016).

$eCO_2$ and global warming increased LAI and GPP in North American boreal forest areas since the 1950s and caused significant positive trends over the Eurasian boreal forest area after the 1980s (**Table 4**). However, due to CV-induced negative effects on tree LAI, no significant trend was found in regional average LAI in boreal areas before the 1980s (**Figure 14**). The LAI and GPP trend enhancement in the boreal forest areas can be attributed to the enhanced $eCO_2$ and global warming effects, accompanied by reduced CV negative effects after the 1980s.

North American subarctic areas had enhanced LAI and GPP positive trends after the 1980s, which were caused by the increase in $eCO_2$ and CV positive effects. All three external forcings had effects on LAI and GPP positive trends in the Eurasian subarctic (**Figure 14** and **Table 4**). Meanwhile, remarkable FRAC changes were found since the 1950s. Our simulation suggested that global warming continually favored shrub invasion into tundra biomes, except in the Eurasian subarctic before the 1980s (**Figure 14**). After the 1980s, this shrubification was enhanced due to increase in the warming

effect. In contrast, $eCO_2$ has promoted tundra expansion and shrub decline over subarctic areas since the 1950s, which mitigated the shrubification. Meanwhile, CV played a role to help tree and C3 grass expansion into subarctic areas, and also altered the shrub and tundra competition. Observations had supported our conclusion that shrub expansion into tundra



ecosystems was linked to climate change (e.g. Myers-Smith et al., 2015), particularly to global warming (e.g. Tape et al., 2006;Elmendorf et al., 2012) and precipitation (e.g. Martin et al., 2017).

Over the Tibetan Plateau, CV dominated the positive LAI and GPP trends since the 1950s, excepted in the case of the GPP increase before the 1980s which was dominated by $eCO_2$. The positive trend enhancements for LAI and GPP after

the 1980s were caused by the impact of both $eCO_2$ and CV. Furthermore, our simulation also suggested that CV favored C3 grasses but harmed tundra biome expansion. However, $eCO_2$ had the opposite effects on those PFTs, in contrast to the CV's.

## 6. Conclusion

This work employs a biophysical-dynamic vegetation model (SSiB4/TRIFFID) to explore the responses of the terrestrial ecosystem to the climate variability and elevated atmospheric $CO_2$ concentration during 1948-2007. The SSiB4/TRIFFID is

evaluated by available satellite data in simulating the land surface carbon fixation, and plant growth and competition. The results show SSiB4/TRIFFID captures the vegetation distribution (spatial correlation larger than 0.87) and temporal variability (regional varying significant temporal correlations of 0.34-0.80 for LAI and 0.45-0.83 for GPP). A series of sensitivity experiments were then conducted to detect the ecosystem trends and attribute the trends to elevated atmospheric $CO_2$ concentration ($eCO_2$), global warming, and climate variability (CV).

In general, $eCO_2$ stimulates vegetation growth mainly in the Equatorial areas, and eastern North America, Western Europe, and Eastern China in the mid-latitudes. The rapidly enhanced global warming after the 1980s contributes positive LAI trends at high latitude, while the GPP change seems less substantial; meanwhile, there were negative trends due to the heat stress in low latitudes. CV dominates the variability of FRAC, LAI and GPP in the semi-humid and semi-arid areas.

The effects of the external drivers on the ecosystem trends manifest distinct spatial and temporal characteristics. For

the global land surface, over 40.2% had a significant LAI trend and over 48.1% had a significant GPP trend since the 1950s. In responding to the climate regime shift during the 1980s, the terrestrial ecosystem had three major changes in different parts of the world after the 1980s. Over 14.2% (LAI) and 11.4% (GPP) of the land surface, primarily located in East Asian monsoon area, West Africa, Central Asia, and Eastern US, had trend sign reversal from negative to positive. In contrast, trend reversal form positive to negative was found in western North America, South America savanna and East Africa,

which accounted for 2.9% (LAI) and 2.7% (GPP) of the land surface. Meanwhile, there were consistent positive trends substantially enhanced in Equatorial rainforest areas, boreal forest areas, South Africa, North Australia, subarctic areas, and the Tibetan Plateau, representing over 11.7% (LAI) and 19.3% (GPP) of the land surface, respectively.

## Data availability

The data for this paper are available at https://drive.google.com/file/d/1CGlO-x-gSXTH8UNB_KWoH_Y53fK-

fcBZ/view?usp=sharing





**Acknowledgements**

This work was supported by NSF Grant AGS-1419526.

**Competing interests**

The authors declare that they have no conflict of interest.

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



| 100-year equilibrium simulation | *Initial condition* → | Real-forcing simulation 1948-2007 |
|---|---|---|
| *Fix $CO_2$ concentration at 1948 level Climatology forcing* | | *Transient $CO_2$ concentration and forcing* |
| • Control experiment<br>• Adjusted little contribution due to large scale disturbance for trees | | • **Control experiment**<br>• **Fixed-$CO_2$**: Fixed $CO_2$ concentration at 1948 level<br>• **Detrend-Temp**: No global warming trend |

**Figure 1. Experimental design flowchart**





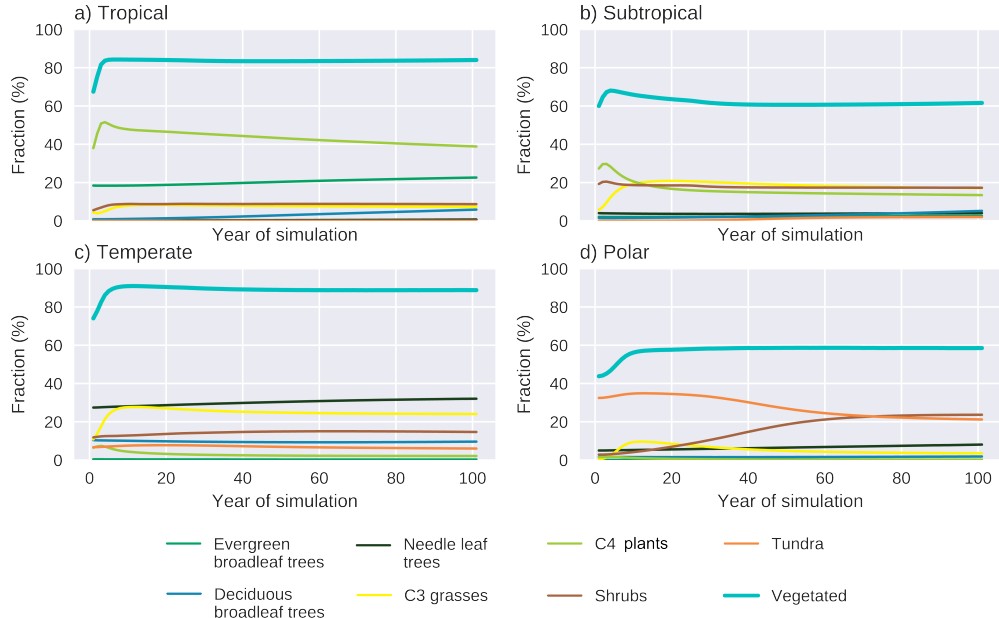

**Figure 2. Fractional coverage of each plant functional type in typical climate zones in the equilibrium experiment**





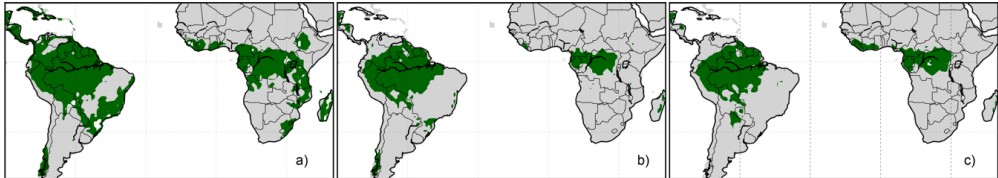

Figure 3. Tree dominated areas of a) unchanged large-scale disturbance experiment, b) parameter updated experiment and c) GLC2000



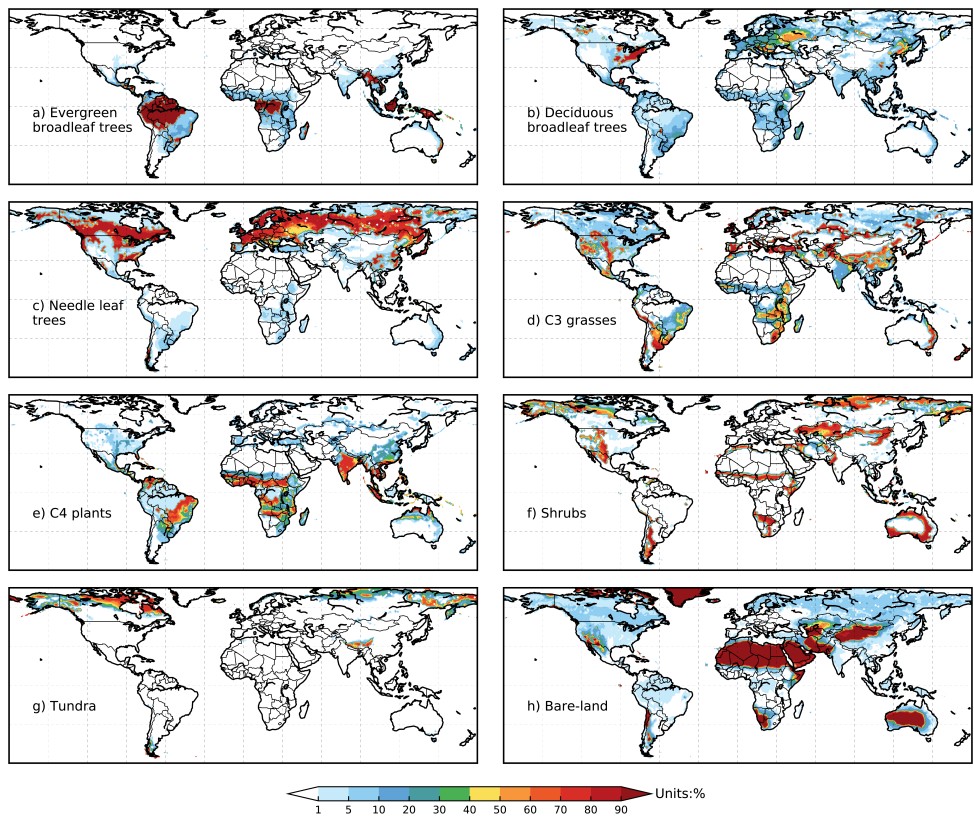

**Figure 4. 1998-2007 average vegetation fractional coverage distribution for different land cover types**





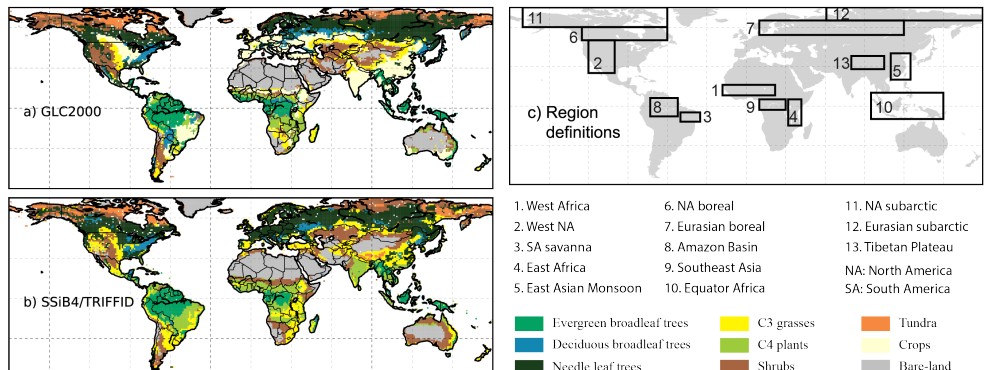

**Figure 5. Dominant vegetation type comparison between a) GLC2000 and b) SSiB4/TRIFFID, and c) Region definitions.**





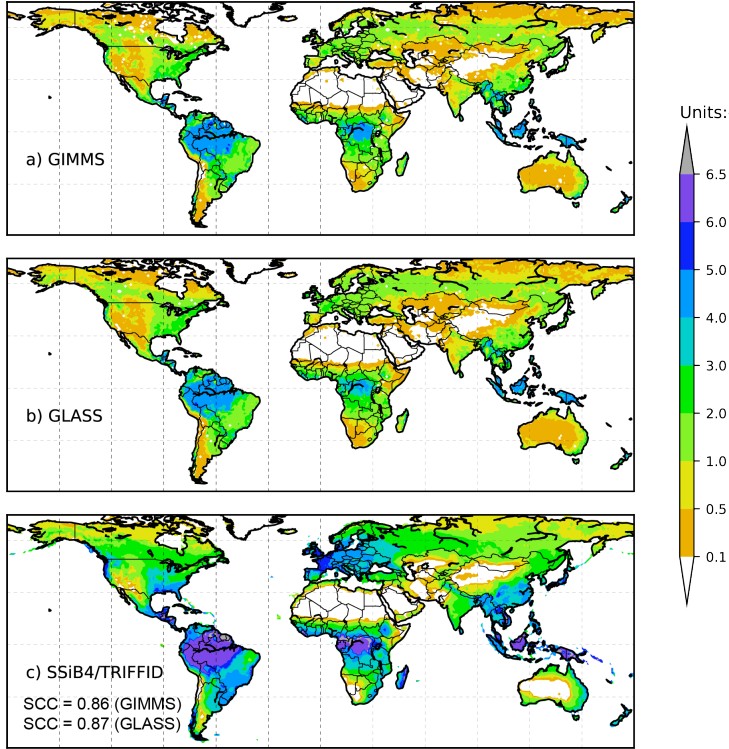

**Figure 6. 1982-2007 average leaf area index comparison for a) GIMMS LAI, b) GLASS LAI, and c) SSiB4/TRIFFID simulated LAI. SCC indicates the spatial correlation coefficient between model simulation and satellite-derived datasets.**





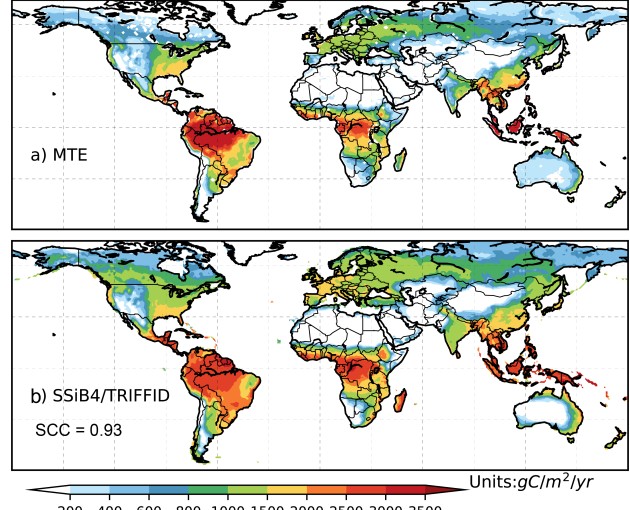

**Figure 7. 1982-2007 average gross primary product comparison for a) MTE GPP, and b) SSiB4/TRIFFID simulated GPP. SCC indicates the spatial correlation coefficient.**





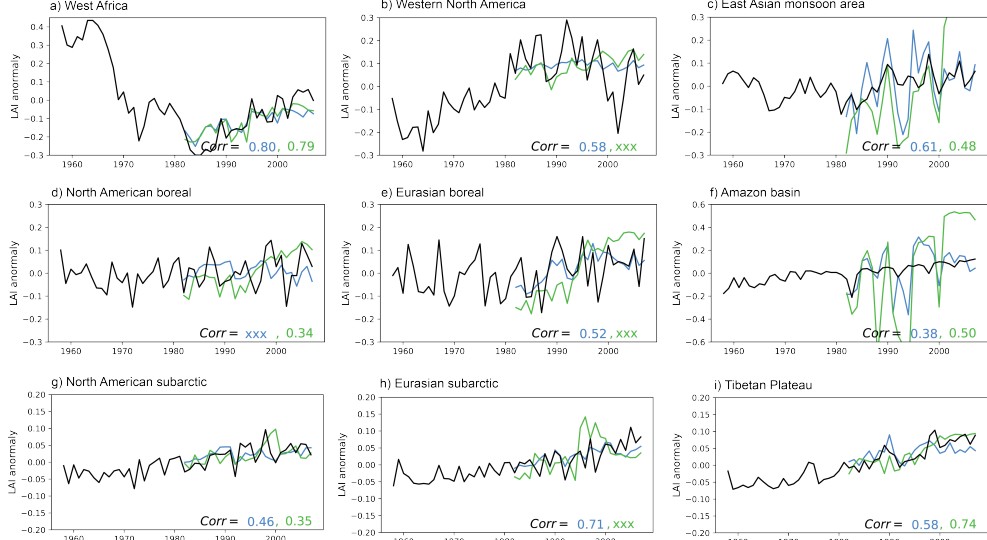

**Figure 8. Comparison of standardized LAI anomalies between simulation and observations for sub-regions. Corr indicates the interannual correlation coefficient simulated LAI against the GIMMS (in blue) and the GLASS (in green). Only significant values (P<0.1) are shown, whereas non-significant values are masked by xxx**



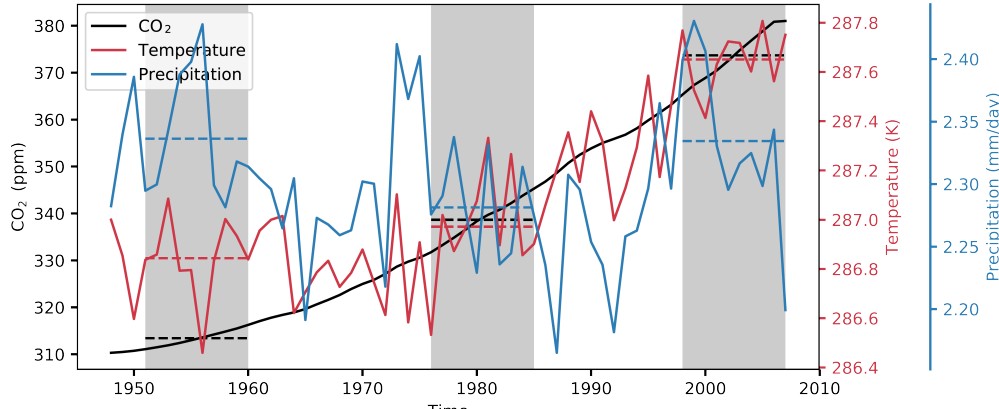

**Figure 9. Global averaged atmosphere CO₂ concentration (black solid line), temperature (red solid line), and precipitation (blue solid line). The dashed lines show the average of 1951-1960, 1977-1986, and 1998-2007, separately**



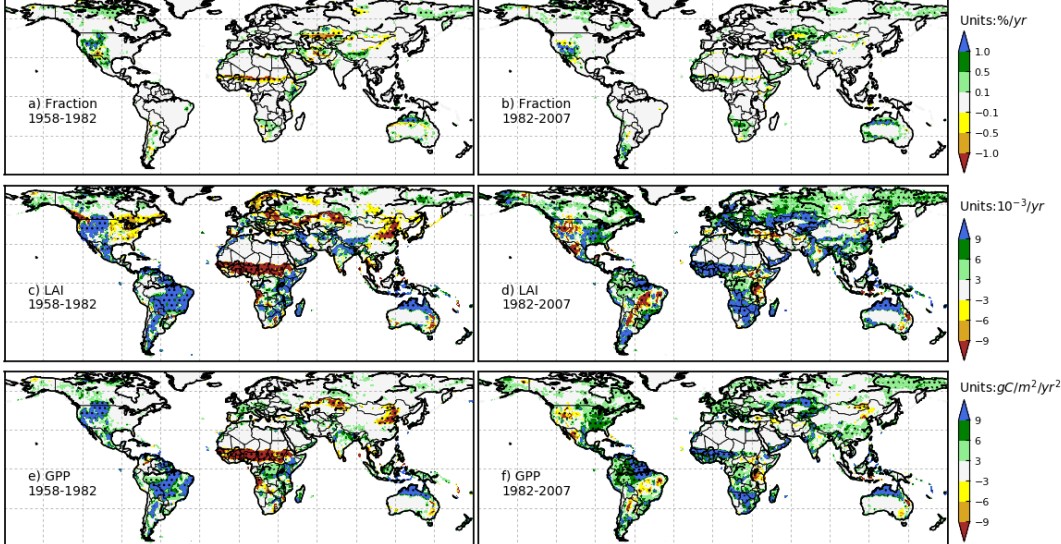

**Figure 10. Trends (shaded) of a) and b) Fractional coverage (units: %/yr), c) and d) LAI (units: 10⁻³/yr), e) and f) GPP (units: gC/m²/yr²). The left three panels are for 1958-1982 and the right three panels are for 1982-2007. The dots indicate the areas with significance level at P< 0.1 (Mann-Kendall test).**





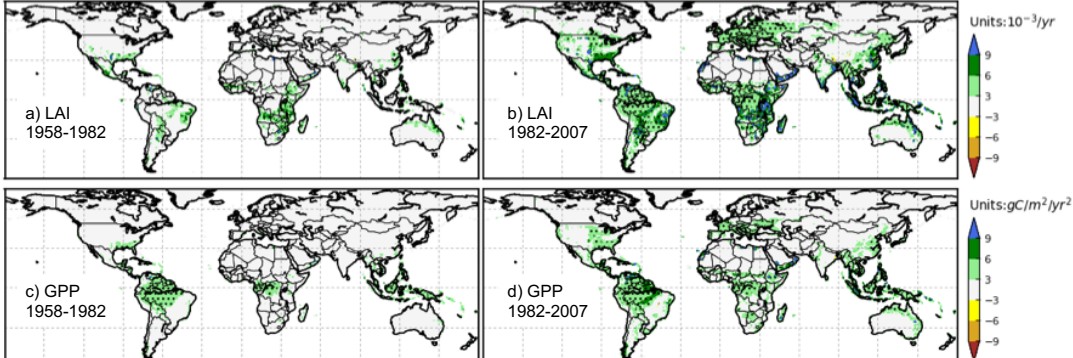

Figure 11. CO$_2$ effect on the trends (shaded) of a) and b) LAI (units: $10^{-3}$/yr), and c) and d) GPP (units: gC/m$^2$/yr$^2$). The left two panels are for 1958-1982 and the right two panels are for 1982-2007. The dots indicate the areas with significance level at P<0.1 (Mann-Kendall test).



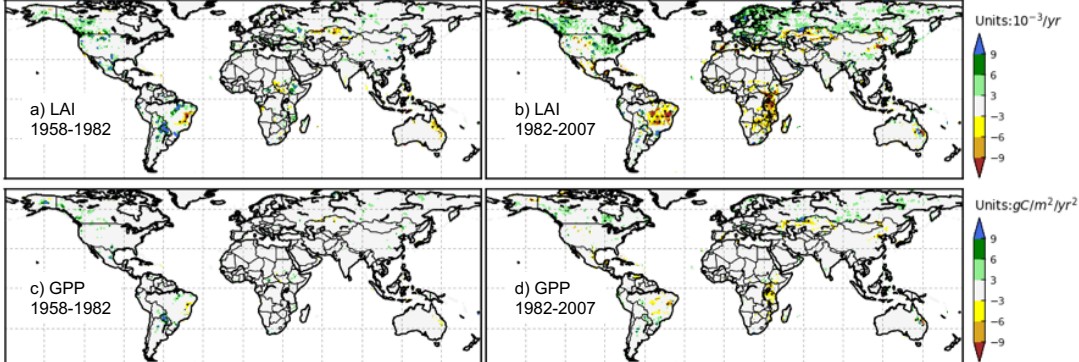

**Figure 12. Same as Figure 11, but for warming effect.**



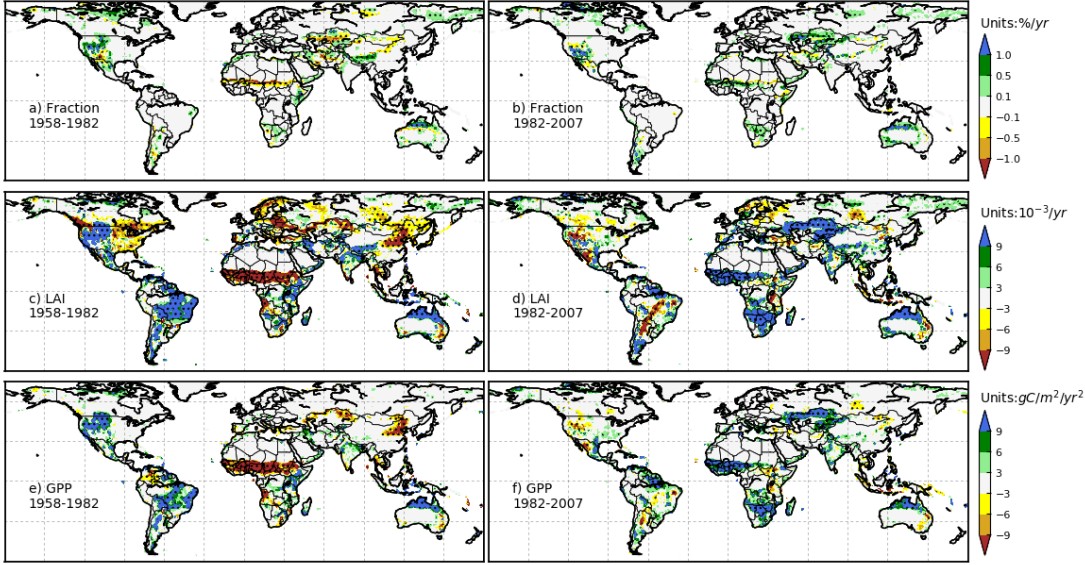

**Figure 13. Same as Figure 11, but for climate viability effect and also including the effects on fractional coverage (units: %/yr)**



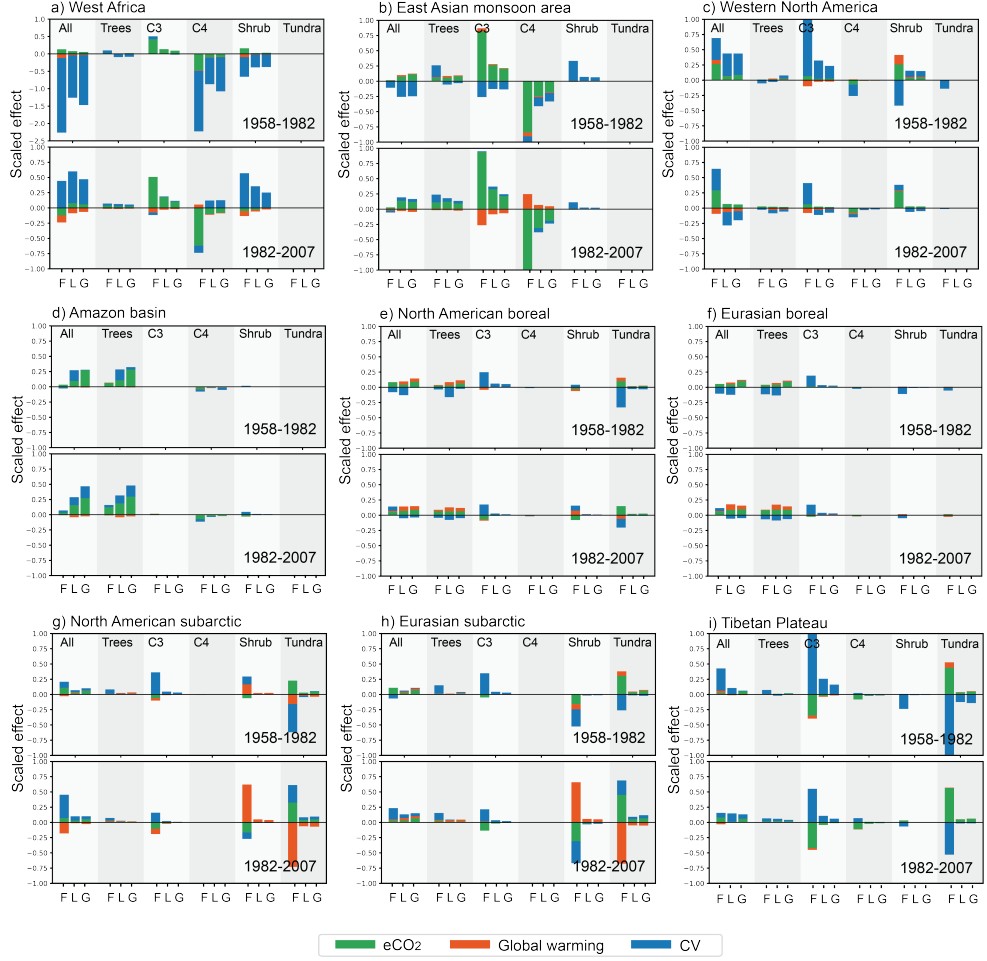

**Figure 14. Contribution of each factor on FRAC (denoted as "F"), LAI (denoted as "L"), and GPP (denoted as "G") trend over sub-regions. The upper panel in each figure is for trends during 1951-1982 and the lower panel is for trends during 1982-2007. The effects of eCO$_2$, global warming and CV are shown in green, red, and blue bars, separately. Each column shows the effects on all PFTs (All), trees, C3 grass (C3), C4 plant (C4), shrub, and tundra, separately. The numbers for FRAC, LAI, and GPP are normalized by dividing the standard deviation of global average in the control experiment.**



**Table 1. Location for study areas**

| Regions | Sub-regions | Location |
|---|---|---|
| **Arid and Semi-Arid Areas** | West Africa | 8°N~16°N, 18°W~22°E |
| | Western North America | 25°N~50°N, 120°W~100°W |
| | South American savanna | 12°S~5°S, 50°W~35°W |
| | East Africa | 15°S~5°N, 32°E~42°E |
| **Monsoon Area** | East Asian monsoon area | 20°N~40°N, 110°E~125°E |
| **Northern Hemisphere boreal areas** | North America boreal | 50°N~60°N, 125°W~60°W |
| | Eurasian boreal | 54°N~65°N, 10°E~120°E |
| **Equator areas** | Amazon basin | 8°S~6°N, 73°W~52°W |
| | Southeast Asia | 10°S~10°N, 95°E~150°E |
| | Equator Africa | 3°S~5°N, 10°E~30°E |
| **Subarctic areas** | North American subarctic | 60°N~75°N, 170°W~60°W |
| | Eurasian subarctic | 65°N~75N, 60°E~180°E |
| **Tibetan Plateau** | Tibetan Plateau | 28°N~38°N, 80°E~105°E |



**Table 2. Statistics for the comparison between SSiB4/TRIFFID simulated and observation-based LAI and GPP**

| Regions | Sub-regions | LAI Mean (m²/m²) | | | LAI TCC | | GPP Mean (gC/m²/yr) | | GPP TCC |
|---|---|---|---|---|---|---|---|---|---|
| | | GIMMS | GLASS | SSiB4/ TRIFFID | GLASS | GLASS | MTE | SSiB4/ TRIFFID | MTE |
| **Arid and Semi-Arid Areas** | West Africa | 0.92 | 0.86 | 1.75 | 0.80** | 0.79** | 759.97 | 970.82 | 0.80** |
| | Western North America | 0.57 | 0.46 | 1.27 | 0.58** | | 404.65 | 520.91 | 0.70** |
| | South American savanna | 1.85 | 1.82 | 3.27 | 0.57** | 0.54** | 1615.28 | 1823.22 | 0.65** |
| | East Africa | 1.49 | 1.38 | 2.95 | 0.46** | | 1188.67 | 1487.25 | 0.63** |
| **Monsoon Area** | East Asian monsoon area | 1.59 | 1.31 | 3.58 | 0.61** | 0.48** | 1402.44 | 1503.18 | 0.51** |
| **Northern Hemisphere boreal areas** | North America boreal | 0.80 | 0.91 | 1.94 | | 0.34* | 544.45 | 867.94 | 0.83** |
| | Eurasian boreal | 1.15 | 1.26 | 2.46 | 0.52** | | 857.16 | 1036.47 | 0.70** |
| **Equator areas** | Amazon basin | 4.18 | 4.28 | 6.15 | 0.38* | 0.50** | 2992.29 | 2820.73 | |
| | Southeast Asia | 4.00 | 3.16 | 4.82 | | 0.38* | 2792.95 | 2628.02 | |
| | Equator Africa | 3.86 | 3.48 | 6.09 | 0.36* | 0.70** | 2535.6 | 2833.9 | |
| **Subarctic areas** | North American subarctic | 0.37 | 0.38 | 0.78 | 0.46** | 0.35* | 256.46 | 438.18 | 0.72** |
| | Eurasian subarctic | 0.35 | 0.44 | 0.92 | 0.71** | | 345.82 | 539.41 | 0.64** |
| **Tibetan Plateau** | Tibetan Plateau | 0.52 | 0.43 | 1.29 | 0.58** | 0.74** | 323.12 | 546.2 | 0.49** |

Note: * indicates the $p < 0.1$ and ** indicates the $p < 0.05$





**Table 3. Linear trend of vegetation FRAC, LAI and GPP changes during P1 (1958-1982) and P2 (1982-2007) in sub-regions (1)**

| Regions | Var. | Total P1 | Total P2 | Trees P1 | Trees P2 | C3 grass P1 | C3 grass P2 | C4 plant P1 | C4 plant P2 | Shrub P1 | Shrub P2 | Tundra P1 | Tundra P2 |
|---|---|---|---|---|---|---|---|---|---|---|---|---|---|
| West Africa | FRAC | -4.14 | | | | **0.96** | **0.99** | -4.33 | -1.71 | -0.96 | **1.1** | | |
| | LAI | -2.58 | **1.37** | | | **0.28** | **0.43** | -1.9 | | -0.79 | **0.8** | | |
| | GPP | -18.54 | **8.02** | -0.84 | **0.86** | **0.92** | **1.9** | -14.09 | | -4.53 | **4.49** | | |
| East Asian Monsoon area | FRAC | | | **0.51** | **0.56** | **1.19** | **1.72** | -2.49 | -2.6 | **0.63** | | | |
| | LAI | -0.34 | **0.43** | | **0.44** | **0.34** | **0.76** | -0.9 | -0.83 | | | | |
| | GPP | -1.61 | **2.39** | **0.84** | **2.17** | **1.06** | **3.51** | -4.35 | -3.73 | **0.84** | **0.44** | | |
| Western North America | FRAC | **1.34** | **1.38** | | | **2.18** | **0.83** | -0.48 | -0.38 | | **0.96** | | |
| | LAI | **0.96** | -0.58 | | | **0.65** | -0.23 | | | **0.34** | | | |
| | GPP | **5.66** | -2.73 | **1.02** | -0.74 | **2.76** | -1.12 | | -0.42 | **1.95** | | | |
| Amazon basin | FRAC | | | | **0.37** | | | | | | | | |
| | LAI | **0.57** | **0.66** | **0.61** | **0.74** | | | | | | | | |
| | GPP | **3.49** | **8.7** | **4.08** | **8.93** | | | -0.67 | -0.39 | | | | |
| North American boreal | FRAC | | **0.36** | | | **0.4** | | | | | | -0.34 | |
| | LAI | | **0.25** | | | | | | | | | | |
| | GPP | **1.77** | **2.2** | **1.17** | **1.41** | **0.64** | | | | | | | **0.33** |
| Eurasian boreal | FRAC | | **0.29** | | | **0.37** | **0.37** | | | | | | |
| | LAI | | **0.33** | | **0.23** | | | | | | | | |
| | GPP | **1.58** | **2.17** | **1.42** | **1.61** | **0.35** | **0.58** | | | | | | |
| North American subarctic | FRAC | **0.35** | **0.69** | **0.16** | **0.18** | **0.51** | | | | **0.46** | **0.88** | -0.76 | -0.29 |
| | LAI | | **0.23** | | | | | | | | | | |
| | GPP | **1.33** | **1.47** | **0.43** | **0.35** | **0.34** | | | | **0.32** | **0.59** | | **0.5** |
| Eurasian subarctic | FRAC | | **0.59** | **0.29** | **0.39** | **0.58** | **0.20** | | | -1.02 | | | |
| | LAI | | **0.35** | | | | | | | | | | |
| | GPP | **1.42** | **2.97** | **0.52** | **0.85** | **0.36** | | | | **0.56** | | **0.69** | **1.31** |
| Tibetan Plateau | FRAC | **0.83** | **0.32** | | | **4.51** | | | | -0.44 | | -3.26 | |
| | LAI | **0.2** | **0.39** | | | **0.48** | | | | | | | |
| | GPP | **0.68** | **2.61** | | **0.78** | **1.9** | **0.98** | | | | | -1.14 | **0.93** |

[1] Only significant values (P<0.1 in Mann-Kendall test) are shown, positive trends are in bold. Numbers are scaled by multiplying 10 for FRAC and 1000 for LAI.





**Table 4. Climate drivers and eCO$_2$ effect on the trends of FRAC, LAI and GPP during P1 (1958-1982) and P2 (1982-2007) in sub-regions regarding to their regional average [1][2]**

| Regions | Var. | Total | | Elevated CO$_2$ concentration | | Global warming | | Climate variability | |
|---|---|---|---|---|---|---|---|---|---|
| | | **P1** | **P2** | **P1** | **P2** | **P1** | **P2** | **P1** | **P2** |
| **West Africa** | FRAC | -0.51 | | | | | | -0.51 | **0.14** |
| | LAI | -1.47 | **0.78** | **0.10** | **0.12** | -0.05 | -0.13 | -1.52 | **0.80** |
| | GPP | -1.91 | **0.83** | **0.07** | **0.12** | -0.05 | -0.13 | -1.94 | **0.83** |
| **East Asian Monsoon area** | FRAC | | | | | | | | |
| | LAI | -0.10 | **0.12** | **0.05** | **0.10** | | | -0.16 | |
| | GPP | -0.11 | **0.16** | **0.10** | **0.16** | | -0.06 | -0.21 | **0.06** |
| **Western North America** | FRAC | **0.18** | **0.18** | **0.07** | **0.10** | | | **0.09** | **0.12** |
| | LAI | **0.75** | -0.45 | **0.11** | **0.14** | | -0.14 | **0.64** | -0.46 |
| | GPP | **1.09** | -0.52 | **0.22** | **0.23** | | -0.19 | **0.88** | -0.56 |
| **Amazon basin** | FRAC | | | | | | | | |
| | LAI | **0.09** | **0.11** | **0.03** | **0.07** | | | **0.06** | **0.06** |
| | GPP | **0.12** | **0.31** | **0.13** | **0.19** | | | | **0.14** |
| **North American boreal** | FRAC | | **0.04** | | | | | | |
| | LAI | | **0.13** | **0.05** | **0.10** | **0.06** | **0.10** | -0.14 | -0.07 |
| | GPP | **0.20** | **0.25** | **0.14** | **0.20** | **0.08** | **0.14** | | -0.08 |
| **Eurasian boreal** | FRAC | | **0.03** | | | | | | |
| | LAI | | **0.13** | | **0.10** | | **0.10** | -0.11 | -0.06 |
| | GPP | **0.15** | **0.21** | **0.13** | **0.18** | | **0.11** | | -0.09 |
| **North American subarctic** | FRAC | **0.05** | **0.09** | | | | -0.06 | | **0.13** |
| | LAI | | **0.30** | **0.07** | **0.10** | | | **0.09** | **0.25** |
| | GPP | **0.30** | **0.34** | **0.17** | **0.24** | | -0.12 | **0.12** | **0.22** |
| **Eurasian subarctic** | FRAC | | **0.06** | | | | | | **0.05** |
| | LAI | | **0.39** | **0.07** | **0.10** | | **0.11** | | **0.18** |
| | GPP | **0.26** | **0.55** | **0.17** | **0.25** | **0.06** | **0.13** | | **0.18** |
| **Tibetan Plateau** | FRAC | **0.12** | **0.04** | | | | | **0.10** | |
| | LAI | **0.16** | **0.31** | | | | | **0.17** | **0.28** |
| | GPP | **0.13** | **0.48** | **0.12** | **0.21** | | | | **0.26** |

[1] percentage trend for FRAC: $Trend_{FRAC}/(grid\ total\ vegetated\ FRAC) * 100\%$
percentage trend for LAI and GPP: $Trend_{var}/(grid\ averaged) * 100\%$, where *var* stands for LAI or GPP. Units for all three variables are %/yr
[2] Only significant values (P<0.1 in Mann-Kendall test) are shown, positive trends are in bold