# Peer review of "Global vegetation variability and its response to elevated CO2, global warming, and climate variability - A study using the offline SSiB4/TRIFFID model and satellite data"

_Earth System Dynamics, 2018_

## Referee Comment (RC1) · Anonymous Referee #1 · 9 Jul 2018

**Summary**

This study provides an attempt to quantify the effects of climate variability and environmental drivers on large-scale trends of carbon fixation and vegetation growth and expansion from 1958-2007 using the SSiB4/TRIFFID dynamic vegetation model. It is clear that a lot of work has gone into the modelling aspect of the study. However, I think that this manuscript requires substantial revisions before I can recommend it for publication in ESD. The structure of the manuscript needs to be revised. I would like you to specify your research questions at the end of the introduction with separate results and discussion sections.

**General comments**

At the moment, this paper feels like a model evaluation paper. There are a lot of results here and it would be nice to see the key results promoted a bit more. Then I would like to see separate results and discussion sections. At the moment, it is a bit mixed up. After the model description section (section 2), I would like section 3 to be the results section, section 4 to be the discussion section and section 5 to be the conclusions section.

The Princeton meteorological dataset is used to drive the SSiB4/TRIFFID model. There is data available from 1948-2010. Why have you not performed the model experiments for this time period?

**Specific comments**

**Abstract**

Change the abstract from 3 paragraphs to 1.

Lines 18-20: You state that more than 40% of the global land area has shows significant trends in LAI and GPP since the 1950s. Is this for the period 1958-2007 or from the 1958-1980s? When you mean positive trends, I assume that is an increase in LAI and GPP and the opposite for negative trends? It is better to explicitly state this.

Lines 22-27: The last paragraph goes straight into which environmental driver affects LAI and GPP the most. Add a line to place the results in context.

**1 Introduction**

Line 3: Remove e.g. when adding references to the end of statements. "...at global and regional scales (e.g. Garcia et al., 2014)" should be "...at global and regional scales (Garcia et al., 2014)". This happens throughout the manuscript. Please remove all occurrences.

Line 4: Change "...by altering fluxes exchanges, energy balance, carbon cycle, etc." to "...by altering the exchange of carbon, water and energy between the atmosphere and land surface".

Line 5: Put references in chronological order. Do this throughout the manuscript.

Lines 5-6: Add a reference for this.

Line 8: Change the definition of LAI from "defined as the one-side leaf area in a unit area" to "defined as the one-sided leaf area per unit ground area"

Line 10: Increasing rate of what? Do you mean a strengthening of the land C sink?

Line 26: Put the word atmospheric before $CO_2$.

Line 27: I would say simulate rather than predict since you are not performing model runs into the future.

Line 28: What are the associated surface characteristics? Give one or two examples (e.g. roughness length, albedo).

Lines 30-31: I would like a reference at the end of this statement. Instead of using the phrase "since the later 1980s", use "towards the end of the 1980s" or "since the second half of the 1980s".

Lines 8-9: Add some references at the end of this sentence.

Lines 9-10: Give an example of some other external forcing.

Line 15: Change "by applying a dynamic global vegetation model" to "by using the SSiB4/TRIFFID (Simplified Simple Biosphere model version 4/Top-down Representation of Interactive Foliage and Flora Including Dynamics) DGVM...". Just use the abbreviation from now on.

Lines 19-24: Remove these lines and add your research questions here.

**2 Model description, observational datasets, and experimental design**
Move section 2.4 (Experimental design) to just after the model description section.

The structure of this section should be:
2.1 Model description
2.2 Experimental design
2.3 Data
2.3.1 Meteorological forcing data
2.3.2 Observation-based data

Line 25: Change 2.1 title to "Model description".

Line 26: Change "...is a biophysically based model incorporating estimates fluxes of radiation..." to "...is a biophysically based model which simulates fluxes of radiation...".

Lines 29-32: Change this sentence to "The TRIFFID DGVM (Cox, 2001) was coupled to SSiB version 4 (Xue et al., 2006) to calculate vegetation dynamics, including relevant land-surface characteristics of vegetation cover and structure."

Line 2: Delete "Some parameters were also updated in this process".

Lines 2-7: I don't think you need this here. Please add a description of how GPP and LAI are calculated in SSiB4.

**2.2 Meteorological forcing data**
In this section, please add more information regarding the Princeton dataset. How was it created? Where did you download it from?
Line 11: " modelling" instead of "modeling".

Lines 13-14: Add units.

**2.3 Observation-based data**
Can you make a section called Data? Add a brief paragraph detailing what datasets were used as input (meteorological, vegetation, soil) to SSiB4/TRIFFID and those used to evaluate the model. Then add the Meteorological forcing data and Observation-based data sections as subsections.

Line 17: Is this the land cover map used by the model or is it used to evaluate the model after spin-up? Where did you download it from? Did you have to do any processing of the land cover map for the study? What is the native resolution of the database?

Line 19: Please explain what S1 is? "SPROT" should be "SPOT".

Line 23: What do you mean by dominance in the GLC2000 dominance map? Is this a map in which the PFT that has the most coverage in each grid box is the dominant PFT?

Line 25: Provide a brief explanation of the differences between the GLC2000 and MODIS land cover map?

Line 26: Do you mean assess and not access?

Lines 27-28: Where did you download GIMMS, GLASS and MODIS LAI datasets from? Can you provide a couple of sentences on the differences between these LAI datasets. Did you have to do any processing of the LAI data?

Lines 1-2: Change from "...remapped to 1-degree spatial resolution and a monthly temporal interval." to "...regridded to 1-degree spatial and monthly temporal resolution."

Lines 3-4: This sentence would be better written as
"SSiB4/TRIFFID GPP was evaluated using the upscaled FLUXNET GPP (hereafter referred to as FLUXNET-MTE) (Jung et al., 2009; Jung et al., 2011)." Also where did you download the data from? Provide more information on how the dataset was created.

Line 7: Change "MTE-GPP data was remapped to 1-degree spatial and a monthly temporal resolution." to "FLUXNET-MTE GPP was regridded to 1-degree spatial and monthly temporal resolution.".

Lines 9-10: Change this sentence to "In this study, SSiB4/TRIFFID was used to simulate the global vegetation distribution and assess the sensitivity of ecosystem trends to climate and eCO2."

Line 10: Remove "For this purpose". "performed" instead of "conducted"

Lines 13-14: Remove the sentence "Meanwhile, the effect of largescale disturbance (LSD) on restricting tree expansion to savanna areas was investigated." Put this in the results/discussion sections.

Line 17: Remove "firstly". What are the multiple biotic variables?

Pages 6-14
**3 Vegetation initial conditions**
This section should now be your results section. Each subsection should have the questions as the heading, so the reader does not have to go back to the introduction again. In each subsection, have a sentence that summarises the main finding. This makes it easier for the reader to understand the key findings.
I suggest the following. Obviously, you should add your own question after each subsection number. I have only added what I think the section should be about.

3 Results
3.1 Evaluating the quasi-equilibrium simulation. Comparison to land cover map. Effect of large-scale disturbance.
3.2 Evaluating GPP/LAI.
3.3 Assessment of vegetation temporal variability.
3.4 External forcing effects on ecosystem trends.

The discussion section (section 4) should have 4 subsections, each discussing the corresponding results from section 3. The Conclusions section will be section 5.

**3.1 Quasi-equilibrium simulation**
Lines 3-16: The first 2 paragraphs could be moved to the discussion section.

Lines 17-25: Move this paragraph to the beginning of this section.

Line 11: Do you mean allocation instead of reallocation?

Line 12: Don't start a sentence with "Figure X shows...". Make a statement regarding the result and reference the figure in brackets at the end of the sentence. Do this throughout the manuscript.

Line 23: Delete the phrase "That being said".

Lines 3-5: Delete the sentence "Detailed comparison of the simulated...".

Line 27: You could reference Figure 4 here.

Line 28: Re-write this sentence as "The spatial correlation coefficient between model and FLUXNET-MTE GPP is 0.93 (P<0.05) (Figure 7)".

Line 30: The standard way to quote global GPP values is in PgC/yr. Remove 1135 gC/m2/yr.

Line 32: Lack of N-limitation.

Line 8: Change "for the model validation" to "for model evaluation".

Line 6: I don't think you need Figure 9 as it is referenced only once. You could add a reference at the end of the sentence instead.

Lines 6-7: State the temperatures as Celsius instead.

Lines 18-19: Remove the information regarding the stipples (dots) on Figure 10 as it is already included in the caption.

Lines 24-25: Reference a figure at the end of this sentence.

Can you include a few sentences on how elevated CO2 affects GPP/LAI in the model as part of the discussion?

Lines 6-8: Remove these lines.

**6. Conclusion**
I would like to see some more discussion in the Conclusions section. It is a bit short. Of the three sensitivity experiments, please state which one is the most important and why?

Line 9: Change "to the climate variability" to " to climate variability".

Lines 10-11: Delete the phrase "The results show". Use "We have shown that the SSiB4/TRIFFID model can simulate the vegetation distribution and temporal variability for the X time period."

Lines 28-30: Is there a University service for making the data available rather than a google drive account? Can you obtain a doi for the data? Also can you specify what data you have made available?

Figure 2
Can you use different colours for the lines as they are difficult to distinguish?

Figure 4
I don't think you need this figure. Can you remove?

Figure 5
Can you include the MODIS dataset since you mention it on Page 4, line 24?

Figure 6
Please plot the differences (model - obs) instead.

Figure 7
Plot the difference instead.

Figure 8
You haven't said what the black line is? Is this SSiB4/TRIFFID?
Change first line of caption to "Comparison of standardized LAI anomalies between simulation and observations for 9 sub-regions."

---

## Referee Comment (RC2) · Anonymous Referee #2 · 9 Jul 2018

I have three main comments about the study by Liu et al. "Global vegetation variability and its response to elevated CO2, global warming, and climate variability - A study using the offline SSiB4/TRIFFID model and satellite data".

First, I fail to see what is new here compare to previously published studies: the current study uses only one model which does not seem to perform better than the TRENDY models used in Zhu et al. (2016) according to the results p. 8 l. 24-27 and p. 9 l. 22-25. As referee #1 mentions, this paper reads like a model evaluation and new scientific insights should be brought forward. If version 5 outperforms version 4 as mentioned p.

8 l. 25-27, the authors should consider using it instead.

Second, why is the study limited to the years 1958-2007? Considering the increasing availability of EO since 2007, extending the study period to nowadays would help address the "global vegetation variability" using satellite data as the title and the introduction (p. 3 l. 17).

Third, there is a lack of consistency between p-values reported, see for example p.9 l. 23 which points to possible cherry-picking from the authors.

Hereafter are some more minor comments:

p. 2 l. 6 Can you support this statement with a reference?

p. 2 l. 9 Leaf area "per unit of ground" area

p. 2 l. 11 Can you quantify the significantly increasing rate to give more context to this statement?

p. 2 l. 25 Consider citing Zhu et al. 2013 as an example of dataset covering the period 1980 to present

p. 2 l. 30 Please cite articles that support this 'general consensus'

p. 3 l. 9 Can you provide a bit more context and examples of these abrupt shifts?

p. 3 l. 15 See my main comment about the study period

p. 3 l. 18 'apportioned' is perhaps more correct than 'attributed'

p. 4 l. 19 Please define SPOT (Satellite Pour l'Observation de la Terre), indicate what type of sensor VEGETATION is and a what resolution these data were available.

p. 4 l. 26 To my knowledge GIMMS is also derived from AVHHR data.

p. 5 l. 3-7 As the study aims to use satellite data, why not using MODIS GPP/NPP (or GIMMS-based NPP from Kolby-Smith et al. 2016).

p. 5 l. 20 onwards Can you summarize the experiments in a table?

p. 6 l. 8 Transient simulations are usually performed from a steady-state obtained under past conditions. Using the average conditions of the period 1948-2007 may reduce the model's sensitivity to the warming that occurred during that period. It would have been better to use the first ∼10 years of driving data for this procedure.

p. 6 l. 15: Is this checked at pixel level, or only globally? Have you checked whether fluxes and initial stocks were at equilibrium? see e.g. Exbrayat et al. (2014) for the importance of initial stocks on transient simulations.

p. 6 l. 22-25 Are these sentences referring to the model or GLC?

p. 7 l. 1 This part is very specific to the model used here. Readers who are not familiar with TRIFFID need a bit of context to understand how the LSD coefficient is used, and the impact of increasing its value ten fold.

p. 7 l. 21 You can also cite Poulter et al. (2014)

p. 8 l. 19 Please clarify whether you are referring to global average LAI.

p. 8 l. 25-27 This statement raises an important question: why do you use model version 4 when you know that model version 5 outperforms it?

p. 9 l. 22-23 Please consider rewording... correlations of 0.35 cannot be described as matching the reference data closely.

p. 10 l 19 Once again p-value...

p. 14 l. 29 I have not been able to access the data using this link, please check.

References for this review

Exbrayat J-F, Pitman AJ, Abramowitz G (2014) Response of microbial decomposition to spin-up explains CMIP5 soil carbon range until 2100. Geosci Model Dev 7:2683–2692. doi: 10.5194/gmd-7-2683-2014
Kolby Smith W, Reed SC, Cleveland CC, et al (2016) Large divergence of satellite and Earth system model estimates of global terrestrial CO2 fertilization. Nat Clim Chang 6:306–310. doi: 10.1038/nclimate2879

Poulter B, Frank D, Ciais P, et al (2014) Contribution of semi-arid ecosystems to inter-annual variability of the global carbon cycle. Nature 509:600–603. doi: 10.1038/nature13376

Zhu Z, Piao S, Myneni RB, et al (2016) Greening of the Earth and its drivers. Nat Clim Chang 6:791–795. doi: 10.1038/nclimate3004
* * *

---

## Author Comment (AC1) · 14 Aug 2018

**Response to Reviewer 1**

**General comments**

At the moment, this paper feels like a model evaluation paper. There are a lot of results here and it would be nice to see the key results promoted a bit more. Then I would like to see separate results and discussion sections. At the moment, it is a bit mixed up. After the model description section (section 2), I would like section 3 to be the results section, section 4 to be the discussion section and section 5 to be the conclusions section.

R: We appreciate Reviewer #1's carefully reading and insightful comments and suggestions. We have made revisions according to the reviewer's comments/suggestions. They really help us to substantially improve the paper. Please find our responses below for each comment.

The purpose of this paper is to investigate the effects of climate regime shift during the 1980s on ecosystem trends, by comparing the contribution of three primary drivers (i.e. atmospheric $CO_2$, global warming, and climate variability) on vegetation cover fraction, LAI and GPP trends, during the periods before and after the 1980s. Studies on this subject show that the contributions of each drivers are model dependent (Beer et al, 2010; Zhu et al, 2016; Huntzinger et al, 2017). Although SSiB and TRIFFID are both well-evaluated models, the coupled version SSiB4/TRIFFID is used for the first time in this study. It is necessary, therefore, to include an evaluation step before its application.

We deemphasize the model evaluation section with fewer paragraphs in our revision. Figure 4 (presenting the vegetation cover fraction) is moved to the appendix. Figure 6 and Figure 7 is modified to show the difference between simulated and satellite-derived LAI and GPP, separately.

The results and discussion sections are separated in the revised manuscript. Key results are emphasized during the discussion and also in the conclusion section.

Meanwhile, the structure is adjusted in the revised manuscript as following:

1. Introduction
2. Model description, experimental design and data
   2.1 Model description
   2.2 Experimental design
   2.3 Data
      2.3.1 Preliminary initial condition
      2.3.2 Meteorological forcing data
      2.3.3 Observation-based data
3. Results
   3.1 Vegetation initial condition

The Princeton meteorological dataset is used to drive the SSiB4/TRIFFID model. There is data available from 1948-2010. Why have you not performed the model experiments for this time period?

R: We have downloaded three versions of Princeton meteorological dataset with the ending year of 2007 (v1x), 2010 (v1) and 2014 (v2.2), respectively. v1 had merged the data v1x plus the data from 2008-2010. However, when we compared the two versions (i.e. v1x and v1), we found that although v1x and v1 are generally consistent before 2007, there was an abrupt shift in some variables (such as wind speed) after 2007 (See Response Figure **1**). To ensure the consistence and minimize the uncertainties associated with the meteorological forcing data, we decided to stop the simulation at 2007. The v2 data, which starts to be available in later 2016, is quite different from the v1 data (Response Figure 1, blue line) for a number of variables. Since by that date we had finished most of our work, and since these few additional years should make no great difference, we have stuck with the v1x data.

[Figure]

Response Figure 1. Comparison between different version of Princeton meteorological datasets over global land (-180º W,180º E, -60º S, 75º N)

**Specific comments**

**Abstract**

Change the abstract from 3 paragraphs to 1.

R: It has been merged to 1 paragraph in the revised manuscript.

Lines 18-20: You state that more than 40% of the global land area has shown significant trends in LAI and GPP since the 1950s. Is this for the period 1958-2007 or from the 1958-1980s? When you mean positive trends, I assume that is an increase in LAI and GPP and the opposite for negative trends? It is better to explicitly state this.

R: Approximately 40% of the global land area shows significant trends in LAI and GPP is for the period 1957-2007. We have clarified it in the revised manuscript.

Yes, the positive trends imply an increase in LAI and GPP and opposite for negative trends. This has been clarified in the revised manuscript.

Lines 22-27: The last paragraph goes straight into which environmental driver affects LAI and GPP the most. Add a line to place the results in context.

R: We have made a number of modifications so that the manuscript flows better.

**Introduction**

Line 3: Remove e.g. when adding references to the end of statements. "...at global and regional scales (e.g. Garcia et al., 2014)" should be "...at global and regional scales (Garcia et al., 2014)". This happens throughout the manuscript. Please remove all occurrences.

R: Removed in the revised manuscript as suggested.

Line 4: Change "...by altering fluxes exchanges, energy balance, carbon cycle, etc." to "...by altering the exchange of carbon, water and energy between the atmosphere and land surface".

R: Done.

Line 5: Put references in chronological order. Do this throughout the manuscript.

R: Done

Lines 5-6: Add a reference for this.

R: References have been added.

Line 8: Change the definition of LAI from "defined as the one-side leaf area in a unit area" to "defined as the one-sided leaf area per unit ground area"

R: Thanks, Corrected in the revised manuscript. Thanks.

Line 10: Increasing rate of what? Do you mean a strengthening of the land C sink?

R: Yes. It is a strengthening of the land carbon sink. We have explained this better in the revised manuscript.

Line 26: Put the word atmospheric before $CO_2$.

R:  Added atmospheric before $CO_2$ in Line 26 and all other occurrences.

Line 27: I would say simulate rather than predict since you are not performing model runs into the future.

R: Changes have been made in Line 27 and other occurrences.

Line 28: What are the associated surface characteristics? Give one or two examples (e.g. roughness length, albedo).

R: Roughness length, albedo, and etc. are the characteristics that we intended to express here. This has been clarified in the revised manuscript.

Lines 30-31: I would like a reference at the end of this statement. Instead of using the phrase "since the later 1980s", use "towards the end of the 1980s" or "since the second half of the 1980s".

R: Changed to "towards the end of the 1980s".  A reference has been added.

Lines 8-9: Add some references at the end of this sentence.

R: References have been added in the revised manuscript.

Lines 9-10: Give an example of some other external forcing.

R: For instance, wind speed and sea surface pressure. This has been added tothe revised manuscript.

Line 15: Change "by applying a dynamic global vegetation model" to "by using the SSiB4/TRIFFID (Simplified Simple Biosphere model version 4/Top-down Representation of Interactive Foliage and Flora Including Dynamics) DGVM...". Just use the abbreviation from now on.

R: Changes have been made in the manuscript according to this suggestion.

5    Lines 19-24: Remove these lines and add your research questions here.

R: Lines 19-24 were removed. The research questions in this paper are: 1) how do the vegetation trends change before and after the 1980s? 2) What is the effect of climate regime shifts during the 1980s on the vegetation trend change? These questions have been added to the manuscript.

**2 Model description, observational datasets, and experimental design**

10   Move section 2.4 (Experimental design) to just after the model description section.

The structure of this section should be:

2.1 Model description

2.2 Experimental design

2.3 Data

15   2.3.1 Meteorological forcing data

2.3.2 Observation-based data

R: We have modified the structure of this section based on the reviewer suggestion.

Line 25: Change 2.1 title to "Model description".

R: Done

20   Line 26: Change "...is a biophysically based model incorporating estimates fluxes of radiation..." to "...is a biophysically based model which simulates fluxes of radiation...".

R: Done.

Lines 29-32: Change this sentence to "The TRIFFID DGVM (Cox, 2001) was coupled to SSiB version 4 (Xue et al., 2006) to calculate vegetation dynamics, including relevant land-surface characteristics of vegetation cover and structure."

25   R: Done.

Line 2: Delete "Some parameters were also updated in this process".

R: Deleted.

Lines 2-7: I don't think you need this here. Please add a description of how GPP and LAI are calculated in SSiB4.

30   R: Lines 2-7 have been removed from the revised manuscript, and the following sentences on how GPP and LAI calculation in SSiB4/TRIFFID are added.

"SSiB estimates net plant photosynthesis assimilation rate, autotrophic respiration and other surface conditions such as canopy temperature and soil moisture for TRIFFID. TRIFFID updates the coverage of a PFT based on the net carbon available to it

and the competition with other PFTs, which is controlled by Lotka-Volterra equations and carbon pool for leaf, wood, and root for each plant functional type. LAI is calculated based on the leaf carbon pool and leaf phenology.

**2.2 Meteorological forcing data**

In this section, please add more information regarding the Princeton dataset. How was it created? Where did you download it from?

R: We have modified this paragraph to including the data source and how it was created.

Line 11: "modelling" instead of "modeling".

R: Done.

Lines 13-14: Add units.

R: Done.

**2.3 Observation-based data**

Can you make a section called Data? Add a brief paragraph detailing what datasets were used as input (meteorological, vegetation, soil) to SSiB4/TRIFFID and those used to evaluate the model. Then add the Meteorological forcing data and Observation-based data sections as subsections.

R: We have rearranged the data information and made a section called Data (section 2.3), which includes three subsections: 2.3.1 introduce preliminary initial condition (covering the vegetation map and soil used as input in the quasi-equilibrium simulation) ; 2.3.2 Meteorological forcing data (covering the Princeton global meteorological dataset used as forcing data for quasi-equilibrium and real-forcing simulations); 2.3.3 Observation-based data (covering GLC2000 and MODIS for vegetation distribution evaluation, GIMMS and GLASS LAI data for LAI evaluation and FLUXNET-GPP for GPP evaluation). The following has been added to summarize the datasets used as input and those used to evaluate the model.

"A SSiB vegetation and soil map is used as the preliminary initial condition for the quasi-equilibrium simulation. 3-hourly meteorological forcing data and its long-term average are used to drive SSiB4/TRIFFID in the real-forcing and quasi-equilibrium simulation, respectively. The satellite-derived products are used to validate and calibrate the model to produce proper vegetation spatial distribution and temporal variability."

Line 17: Is this the land cover map used by the model or is it used to evaluate the model after spin-up? Where did you download it from? Did you have to do any processing of the land cover map for the study? What is the native resolution of the database?

R: The global land cover map was used to evaluation the model results driven by 1948-2007 meteorological forcing data. It was download from http://forobs.jrc.ec.europa.eu/products/glc2000/glc2000.php. This dataset consists of a global map with one legend, as well as regional maps with separate legends containing more detailed classification for certain regions. We used the 1000 m resolution reginal products to generate land cover fraction map by counting the percentage of each PFT in a 1-degree grid. Then the regional fraction maps were merged to obtain a global land cover fraction map. Furthermore, a land cover map for dominant type at 1-degree resolution was generated based on the vegetation type having the largest coverage in a 1-degree grid box.

Line 19: Please explain what S1 is? "SPROT" should be "SPOT".

R: "S1" stands for SPOT-VEGETATION standard product S1: daily maximum of NDVI composite of spectral reflectance at the top-of-canopy. We have modified this sentence to "The Global Land Cover (GLC) database for the year 2000 (Bartholome et al., 2002) used the data from Satellite Poul l'Observation de la Terre at the spatial resolution about 1000 m" .

Line 23: What do you mean by dominance in the GLC2000 dominance map? Is this a map in which the PFT that has the most coverage in each grid box is the dominant PFT? Line 25: Provide a brief explanation of the differences between the GLC2000 and MODIS land cover map?

R: As mentioned in response to the previous question "Did you have to do any processing of the land cover map for the study?", the high resolution (1000 m) vegetation type product was converted to vegetation fraction map by counting the percentage of each PFT in a 1-degree grid. Then the dominant vegetation map (at 1-degree resolution) was generated, by assigning the type with maxim fraction cover in each grid of the fraction map.

GLC2000 and MODIS are derived from different sensors on-board different satellite and in different classification system. GLC2000 is based on the daily data from VEGETATION sensor on-board Satellite Poul l'Observation de la Terre (SPOT) 4, while MODIS land cover map is based on the Moderate Resolution Imaging Spectroradiometer (MODIS) on board the Terra-1 satellite. MODIS produces the vegetation every year which another product only has GLC2000 and GLC 2014. We feel GLC data may have better quality control.

Line 26: Do you mean assess and not access?

R: Thanks. This typo has been corrected.

Lines 27-28: Where did you download GIMMS, GLASS and MODIS LAI datasets from? Can you provide a couple of sentences on the differences between these LAI datasets. Did you have to do any processing of the LAI data?

R: The Global Inventory Modeling and Mapping Studies (GIMMS) LAI (refer to LAI3g, the third generation) was downloaded from https://ecocast.arc.nasa.gov/data/pub/gimms/. A neural network algorithm was trained to using the AVHRR GIMMS NDVI3g (covering the period July 1981 to December 2011) and best-quality Terra MOIDS LAI (covering the period 2000 to 2009) for the overlapping period 2000-2009. Then the trained neural network algorithm was used to generate corresponding LAI dataset at 15-day temporal resolution and 1/12-degree spatial resolution for the period from July 1981 to December 2011.

The Global Land Surface Satellite (GLASS) LAI was download from http://www.bnu-datacenter.com/. The GLASS LAI was generated from AVHRR reflectance (1982-1999) and MODIS reflectance (2000-2012). The GLASS LAI provides observations at 8-day temporal resolution and 1 km spatial resolution for the period from 1982 to 2012.

The MODIS LAI includes products derived from Terra and Aqua platform and product derived from the combination of the two platforms. The MODIS products are at 8-day temporal resolution and 500 m spatial resolution.

GIMMS LAI and GLASS LAI are used to evaluate the spatial distribution and temporal variability of model simulation. We didn't use MODIS LAI for the comparison directly. GIMMS LAI and GLASS LAI products are averaged to monthly mean, and then regridded to 1-degree spatial resolution.

Above information is included in the revised manuscript.

Page 5 Lines 1-2: Change from "...remapped to 1-degree spatial resolution and a monthly temporal interval." to "...regridded to 1-degree spatial and monthly temporal resolution."

R: Done

Lines 3-4: This sentence would be better written as "SSiB4/TRIFFID GPP was evaluated using the upscaled FLUXNET GPP (hereafter referred to as FLUXNET-MTE) (Jung et al., 2009; Jung et al., 2011)." Also where did you download the data from? Provide more information on how the dataset was created.

R: The sentence is re-written and additional information is provided. The FLUXNET-MTE GPP was downloaded from https://www.bgc-jena.mpg.de/geodb/projects/Data.php. The FLUXNET observations of carbon dioxide flux were upscaled to the global scale using the machine learning technique, model tree ensembles (MTE), which was trained to predict site-level GPP based on remote sensing indices, climate and meteorological data, and information on land use. This data set provides global monthly mean GPP at 0.5-degree spatial resolution for the period from 1982 to 2011. The FLUXNET-MTE GPP was regridded to 1-degree spatial resolution.

Line 7: Change "MTE-GPP data was remapped to 1-degree spatial and a monthly temporal resolution." to "FLUXNET-MTE GPP was regridded to 1-degree spatial and monthly temporal resolution.".

R: Done.

Lines 9-10: Change this sentence to "In this study, SSiB4/TRIFFID was used to simulate the global vegetation distribution and assess the sensitivity of ecosystem trends to climate and eCO2."

R: Done.

Line 10: Remove "For this purpose". "performed" instead of "conducted"

R: Done.

Lines 13-14: Remove the sentence "Meanwhile, the effect of largescale disturbance (LSD) on restricting tree expansion to savanna areas was investigated." Put this in the results/discussion sections.

R: Moved to the results/discussion sections.

Line 17: Remove "firstly". What are the multiple biotic variables?

R: "firstly" is removed. The multiple biotic variables stand for vegetation coverage, LAI and GPP. It has been replaced by "vegetation coverage, LAI and GPP" in the revised manuscript.

Pages 6-14

**3 Vegetation initial conditions**

This section should now be your results section. Each subsection should have the questions as the heading, so the reader does not have to go back to the introduction again. In each subsection, have a sentence that summarizes the main finding. This makes it easier for the reader to understand the key findings.

I suggest the following. Obviously, you should add your own question after each subsection number. I have only added what I think the section should be about.

3 Results

3.1 Evaluating the quasi-equilibrium simulation. Comparison to land cover map. Effect of large-scale disturbance.

3.2 Evaluating GPP/LAI.

3.3 Assessment of vegetation temporal variability.

3.4 External forcing effects on ecosystem trends.

R: The structure of results section has been modified accordingly in the revised manuscript.

3 Results

3.1 Vegetation initial condition

3.2 Evaluation of simulated ecosystem

3.3 Simulated vegetation temporal variability and trend changes

3.4 Attribution of three external forcings on ecosystem trends

We will add a sentence as the beginning of each subsection to point out the main content in this section.

The discussion section (section 4) should have 4 subsections, each discussing the corresponding results from section 3. The Conclusions section will be section 5.

R: The discussions currently in Section 3 have been moved to the discussion section, which now consist of 4 subsections.

**3.1 Quasi-equilibrium simulation**

Lines 3-16: The first 2 paragraphs could be moved to the discussion section.

R: Done.

Lines 17-25: Move this paragraph to the beginning of this section.

R: Done.

Line 11: Do you mean allocation instead of reallocation? Line 12: Don't start a sentence with "Figure X shows...". Make a statement regarding the result and reference the figure in brackets at the end of the sentence. Do this throughout the manuscript.

R: Yes, it is "allocation". This has been corrected in the revised manuscript.

Thanks for pointing out, we have gone through manuscript to reduce use of this phrase.

Line 23: Delete the phrase "That being said".

R: Deleted.

Page 7 Lines 3-5: Delete the sentence "Detailed comparison of the simulated...".

R: Deleted.

Line 27: You could reference Figure 4 here.

R: Done

Page 8 Line 28: Re-write this sentence as "The spatial correlation coefficient between model and FLUXNET-MTE GPP is 0.93 (P<0.05) (Figure 7)".

R: Modified as suggested.

Line 30: The standard way to quote global GPP values is in PgC/yr. Remove 1135 gC/m2/yr.

R: Modified.

Line 32: Lack of N-limitation.

R: Corrected.

Page 9 Line 8: Change "for the model validation" to "for model evaluation".

5 R: Changed.

Page 10 Line 6: I don't think you need Figure 9 as it is referenced only once. You could add a reference at the end of the sentence instead.

R: Figure 9 has been moved to the appendix, and relative references were added.

Lines 6-7: State the temperatures as Celsius instead.

10 R: Done

Lines 18-19: Remove the information regarding the stipples (dots) on Figure 10 as it is already included in the caption.

R: Done

Lines 24-25: Reference a figure at the end of this sentence.

R: This sentence refers to Figure 10 (Figure 8 in the revised manuscript). It is added in the revised manuscript.

15 Page 11 Can you include a few sentences on how elevated CO2 affects GPP/LAI in the model as part of the discussion?

R: Yes. SSiB uses Collatz et al model to calculate photosynthesis process (Zhan et al, 2003) which is given the following equations:

$$A_n = \frac{g_b}{1.4} \frac{C_a - C_s}{p}$$

$$GPP = A_n + R_d$$

20 where $A_n$ is the net $CO_2$ assimilation, $g_b$ is stomatal conductance to latent and sensible heat transfer, $C_a$ is the atmospheric $CO_2$ concentration, $C_s$ is the $CO_2$ concentration at leaf surface, $p$ is the air pressure, and is $R_d$ the dark respiration rate of the canopy. Based on above equations, increase in $C_a$ leads to a larger $A_n$, then a larger GPP.

Vegetation in TRIFFID is presented as leaf, root, and wood carbon, and for each a corresponding carbon pool is updated the based on the net carbon available to it and the competition with other PFTs, which is controlled by Lotka-Volterra

25 equations. LAI is calculated based on the leaf carbon pool and leaf phenology. Larger GPP leads more carbon goes in to the vegetation carbon pool in leaf, root and wood, then larger LAI.

A summary of these explanations have been added to the discussion section.

Page 12 Lines 6-8: Remove these lines.

R: Removed.

30 Page 14

**6. Conclusion**

I would like to see some more discussion in the Conclusions section. It is a bit short. Of the three sensitivity experiments, please state which one is the most important and why?

R: This is a valid point, we agree that the conclusion concerning driver contributions is a bit short. We have included additional discussion in the revised manuscript.

Line 9: Change "to the climate variability" to " to climate variability".

R: Changed.

5   Lines 10-11: Delete the phrase "The results show". Use "We have shown that the SSiB4/TRIFFID model can simulate the vegetation distribution and temporal variability for the X time period."

R: Done.

Lines 28-30: Is there a University service for making the data available rather than a google drive account? Can you obtain a doi for the data? Also can you specify what data you have made available?

10   R: We have uploaded the available data to a University server and will indicate the available data in "data availability" section.. We will request a doi from your journal for the data when we submit the revision.

Figure 2 Can you use different colors for the lines as they are difficult to distinguish? Figure 4 I don't think you need this figure. Can you remove?

R: We have redrawn Figure 2 with different colors to make it clearer. Figure 4 has been moved to the appendix.

15   Figure 5 Can you include the MODIS dataset since you mention it on Page 4, line 24?

R: We have added a vegetation map for MODIS.

Figure 6 Please plot the differences (model - obs) instead.

R: This figure has been replaced.

Figure 7 Plot the difference instead.

20   R: We have redrawn this figure to show the difference.

Figure 8 You haven't said what the black line is? Is this SSiB4/TRIFFID?

R: Yes, it is SSiB4/TRIFFID. It has been added to the revised manuscript.

Change first line of caption to "Comparison of standardized LAI anomalies between simulation and observations for 9 sub-regions."

25   R: Done.

---

## Author Response (AR1)

**Response to Reviewer 1**

**General comments**

At the moment, this paper feels like a model evaluation paper. There are a lot of results here and it would be nice to see the key results promoted a bit more. Then I would like to see separate results and discussion sections. At the moment, it is a bit mixed up. After the model description section (section 2), I would like section 3 to be the results section, section 4 to be the discussion section and section 5 to be the conclusions section.

R: We appreciate Reviewer #1's carefully reading and insightful comments and suggestions. We have made revisions according to the reviewer's comments/suggestions. They really help us to substantially improve the paper. Please find our responses below for each comment. Please note that the reviewer's comments/suggestions are in black, while responses are in blue. The page and line numbers in the response are regarding to the clean version "manuscript.pdf".

The purpose of this paper is to investigate the effects of climate regime shift during the 1980s on ecosystem trends, by comparing the contribution of three primary drivers (i.e. atmospheric $CO_2$, global warming, and climate variability) on vegetation cover fraction, LAI and GPP trends, during the periods before and after the 1980s. Studies on this subject show that the contributions of each drivers are model dependent (Beer et al, 2010; Zhu et al, 2016; Huntzinger et al, 2017). Although SSiB and TRIFFID are both well-evaluated models, the coupled version SSiB4/TRIFFID is used for the first time in this study. It is necessary, therefore, to include an evaluation step before its application.

We deemphasize the model evaluation section with fewer paragraphs in our revision. Figure 4 (presenting the vegetation cover fraction) is moved to the appendix. Figure 6 and Figure 7 is modified to show the difference between simulated and satellite-derived LAI and GPP, separately.

The results and discussion sections are separated in the revised manuscript. Key results are emphasized during the discussion and also in the conclusion section.

Meanwhile, the structure is adjusted in the revised manuscript as following:

    1. Introduction

4. Discussion

5. Conclusion

The Princeton meteorological dataset is used to drive the SSiB4/TRIFFID model. There is data available from 1948-2010. Why have you not performed the model experiments for this time period?

R: We have downloaded three versions of Princeton meteorological dataset with the ending year of 2007 (v1x), 2010 (v1) and 2014 (v2.2), respectively. v1 had merged the data v1x plus the data from 2008-2010. However, when we compared the two versions (i.e. v1x and v1), we found that although v1x and v1 are generally consistent before 2007, there was an abrupt shift in some variables (such as wind speed) after 2007 (See Response Figure 2). To ensure the consistence and minimize the uncertainties associated with the meteorological forcing data, we decided to stop the simulation at 2007. The v2 data, which starts to be available in later 2016, is quite different from the v1 data (Response Figure 1, blue line) for a number of variables. Since by that date we had finished most of our work, and since these few additional years should make no great difference, we have stuck with the v1x data.

[Figure]

Response Figure 1. Comparison between different version of Princeton meteorological datasets over global land (-180° W,180° E, -60° S, 75° N)

**Specific comments**

**Abstract**

Change the abstract from 3 paragraphs to 1.

5   R: It has been merged to 1 paragraph in the revised manuscript.

Lines 18-20: You state that more than 40% of the global land area has shown significant trends in LAI and GPP since the 1950s. Is this for the period 1958-2007 or from the 1958-1980s? When you mean positive trends, I assume that is an increase in LAI and GPP and the opposite for negative trends? It is better to explicitly state this.

10  R: Approximately 40% of the global land area shows significant trends in LAI and GPP is for the period 1957-2007.  We have clarified it in the revised manuscript (page 2 line 10).

Yes, the positive trends imply an increase in LAI and GPP and opposite for negative trends. This has been clarified in the revised manuscript (page 2 line 10).

Lines 22-27: The last paragraph goes straight into which environmental driver affects LAI and GPP the

15  most. Add a line to place the results in context.

R:  We have made a number of modifications so that the manuscript flows better (page 2 line 15).

**Introduction**

Line 3: Remove e.g. when adding references to the end of statements. "...at global and regional scales

20  (e.g. Garcia et al., 2014)" should be "...at global and regional scales (Garcia et al., 2014)". This happens throughout the manuscript. Please remove all occurrences.

R: Removed in the revised manuscript as suggested (page 3 line 4 and other occurrences).

Line 4: Change "...by altering fluxes exchanges, energy balance, carbon cycle, etc." to "...by altering the exchange of carbon, water and energy between the atmosphere and land surface".

25  R: Done in page 3 line 5.

Line 5: Put references in chronological order. Do this throughout the manuscript.

R: Done in page 3 lines 5-6 and other occurrences.

Lines 5-6: Add a reference for this.

R: References have been added in page 3 lines 7-8.

Line 8: Change the definition of LAI from "defined as the one-side leaf area in a unit area" to "defined as the one-sided leaf area per unit ground area"

R: Thanks, Corrected in the revised manuscript in page 3 lines 10-11. Thanks.

Line 10: Increasing rate of what? Do you mean a strengthening of the land C sink?

R: Yes. It is a strengthening of the land carbon sink. We have explained this better in the revised manuscript in page 3 line 13.

Line 26: Put the word atmospheric before $CO_2$.

R: Added atmospheric before $CO_2$ in page 4 line 4 and all other occurrences.

Line 27: I would say simulate rather than predict since you are not performing model runs into the future.

R: Changes have been made in page 4 line 4 and other occurrences.

Line 28: What are the associated surface characteristics? Give one or two examples (e.g. roughness length, albedo).

R: Roughness length, albedo, PFT distribution, LAI and etc. are the characteristics that we intended to express here. This has been clarified in the revised manuscript in page 4 line 4.

Lines 30-31: I would like a reference at the end of this statement. Instead of using the phrase "since the later 1980s", use "towards the end of the 1980s" or "since the second half of the 1980s".

R: Changed to "towards the end of the 1980s". A reference has been added. Please find in page 4 lines 10-11.

Lines 8-9: Add some references at the end of this sentence.

R: References have been added in the revised manuscript in page 4 line 23.

Lines 9-10: Give an example of some other external forcing.

R: For instance, wind speed and sea surface pressure. This has been added to the revised manuscript in page 4 line 25.

Line 15: Change "by applying a dynamic global vegetation model" to "by using the SSiB4/TRIFFID (Simplified Simple Biosphere model version 4/Top-down Representation of Interactive Foliage and Flora Including Dynamics) DGVM...". Just use the abbreviation from now on.

R: Changes have been made in the manuscript according to this suggestion (page 5 lines 3-5).

5   Lines 19-24: Remove these lines and add your research questions here.

R: Lines were removed. The research questions in this paper are: 1) how do the vegetation trends change before and after the 1980s? 2) What is the effect of climate regime shifts during the 1980s on the vegetation trend change? These questions have been added to the manuscript in page 5 lines 8-10.

**2 Model description, observational datasets, and experimental design**

10   Move section 2.4 (Experimental design) to just after the model description section.

The structure of this section should be:

2.1 Model description

2.2 Experimental design

2.3 Data

15   2.3.1 Meteorological forcing data

2.3.2 Observation-based data

R: We have modified the structure of this section based on the reviewer suggestion.

Line 25: Change 2.1 title to "Model description".

R: Done (page 5 line 12).

20   Line 26: Change "...is a biophysically based model incorporating estimates fluxes of radiation..." to "...is a biophysically based model which simulates fluxes of radiation...".

R: Done (page 5 lines 13-15).

Lines 29-32: Change this sentence to "The TRIFFID DGVM (Cox, 2001) was coupled to SSiB version 4 (Xue et al., 2006) to calculate vegetation dynamics, including relevant land-surface characteristics of

25   vegetation cover and structure."

R: Done (page 5 lines 16-18).

Line 2: Delete "Some parameters were also updated in this process".

R: Deleted.

Lines 2-7: I don't think you need this here. Please add a description of how GPP and LAI are calculated in SSiB4.

R: Lines have been removed from the revised manuscript, and the following sentences on how GPP and LAI calculation in SSiB4/TRIFFID are added in page 5 lines 21-26.

"SSiB4 estimates net plant photosynthesis assimilation rate, autotrophic respiration and other surface conditions such as canopy temperature and soil moisture for TRIFFID. TRIFFID updates the coverage of a PFT based on the net carbon available to it and the competition with other PFTs, which is controlled by the Lotka-Volterra equations. Vegetation is described by leaf, wood, and root with associating carbon pools. Leaf phenology is simulated as a function of canopy temperature and soil moisture."

**2.2 Meteorological forcing data**

In this section, please add more information regarding the Princeton dataset. How was it created? Where did you download it from?

R: We have modified this paragraph to including the data source and how it was created. Please find it in page 7 lines 14-22.

Line 11: "modelling" instead of "modeling".

R: Done in page 7 line 14.

Lines 13-14: Add units.

R: Done in page 7 lines 19-21.

**2.3 Observation-based data**

Can you make a section called Data? Add a brief paragraph detailing what datasets were used as input (meteorological, vegetation, soil) to SSiB4/TRIFFID and those used to evaluate the model. Then add the Meteorological forcing data and Observation-based data sections as subsections.

R: We have rearranged the data information and made a section called Data (section 2.3), which includes three subsections: 2.3.1 Initial condition for equilibrium simulation (covering the vegetation map and soil used as input in the quasi-equilibrium simulation) ; 2.3.2 Meteorological forcing data (covering the Princeton global meteorological dataset used as forcing data for quasi-equilibrium and

real-forcing simulations); 2.3.3 Observation-based data (covering GLC2000 and MODIS for vegetation distribution evaluation, GIMMS and GLASS LAI data for LAI evaluation and FLUXNET-GPP for GPP evaluation). The following has been added to summarize the datasets used as input and those used to evaluate the model (page 7 lines 2-6).

5 "A SSiB vegetation and soil map is used as the preliminary initial condition for the quasi-equilibrium simulation. A 3-hourly meteorological forcing data from 1948 through 2007 (Sheffield et al., 2006) is used for this study. The observation-based LAI and GPP products (Zhu et al., 2013; Xiao et al., 2014; Jung et al., 2009) are used to validate and calibrate the model to produce proper vegetation spatial distribution and temporal variability."

10 Line 17: Is this the land cover map used by the model or is it used to evaluate the model after spin-up? Where did you download it from? Did you have to do any processing of the land cover map for the study? What is the native resolution of the database?

R: The global land cover map was used to evaluation the model results driven by 1948-2007 meteorological forcing data. It was download from 15 http://forobs.jrc.ec.europa.eu/products/glc2000/glc2000.php. This dataset consists of a global map with one legend, as well as regional maps with separate legends containing more detailed classification for certain regions. We used the 1000 m resolution reginal products to generate land cover fraction map by counting the percentage of each PFT in a 1-degree grid. Then the regional fraction maps were merged to obtain a global land cover fraction map. Furthermore, a land cover map for dominant type at 1-degree 20 resolution was generated based on the vegetation type having the largest coverage in a 1-degree grid box. The paragraph introducing vegetation cover data has been rewritten in page 7 lines 24-26 and page 8 lines 1-15.

Line 19: Please explain what S1 is? "SPROT" should be "SPOT".

R: "S1" stands for SPOT-VEGETATION standard product S1: daily maximum of NDVI composite of 25 spectral reflectance at the top-of-canopy. We have modified this sentence to "The Global Land Cover (GLC) database for the year 2000 (Bartholome et al., 2002) used the data from Satellite Poul l'Observation de la Terre at the spatial resolution about 1 km" in page 7 lines 25-26 and page 8 line 1.

Line 23: What do you mean by dominance in the GLC2000 dominance map? Is this a map in which the PFT that has the most coverage in each grid box is the dominant PFT? Line 25: Provide a brief explanation of the differences between the GLC2000 and MODIS land cover map?

R: As mentioned in response to the previous question "Did you have to do any processing of the land cover map for the study?", the high resolution (1000 m) vegetation type product was converted to vegetation fraction map by counting the percentage of each PFT in a 1-degree grid. Then the dominant vegetation map (at 1-degree resolution) was generated, by assigning the type with maxim fraction cover in each grid of the fraction map.

GLC2000 and MODIS are derived from different sensors on-board different satellite and in different classification system. GLC2000 is based on the daily data from VEGETATION sensor on-board Satellite Poul l'Observation de la Terre (SPOT) 4, while MODIS land cover map is based on the Moderate Resolution Imaging Spectroradiometer (MODIS) on board the Terra-1 satellite. MODIS produces the vegetation every year which another product only has GLC2000 and GLC 2014. We feel GLC data may have better quality control.

Line 26: Do you mean assess and not access?

R: Thanks. This typo has been corrected in page 8 line 16.

Lines 27-28: Where did you download GIMMS, GLASS and MODIS LAI datasets from? Can you provide a couple of sentences on the differences between these LAI datasets? Did you have to do any processing of the LAI data?

R: The Global Inventory Modeling and Mapping Studies (GIMMS) LAI (refer to LAI3g, the third generation) was downloaded from https://ecocast.arc.nasa.gov/data/pub/gimms/. A neural network algorithm was trained to using the AVHRR GIMMS NDVI3g (covering the period July 1981 to December 2011) and best-quality Terra MOIDS LAI (covering the period 2000 to 2009) for the overlapping period 2000-2009. Then the trained neural network algorithm was used to generate corresponding LAI dataset at 15-day temporal resolution and 1/12-degree spatial resolution for the period from July 1981 to December 2011.

The Global Land Surface Satellite (GLASS) LAI was download from http://www.bnu-datacenter.com/. The GLASS LAI was generated from AVHRR reflectance (1982-1999) and MODIS

reflectance (2000-2012). The GLASS LAI provides observations at 8-day temporal resolution and 1 km spatial resolution for the period from 1982 to 2012.

The MODIS LAI includes products derived from Terra and Aqua platform and product derived from the combination of the two platforms.  The MODIS products are at 8-day temporal resolution and 500 m spatial resolution.

GIMMS LAI and GLASS LAI are used to evaluate the spatial distribution and temporal variability of model simulation. We didn't use MODIS LAI for the comparison directly. GIMMS LAI and GLASS LAI products are averaged to monthly mean, and then regridded to 1-degree spatial resolution.

Above information is included in the revised manuscript in page 8 lines 16-28 and page 9 lines 1-3.

Page 5 Lines 1-2: Change from "...remapped to 1-degree spatial resolution and a monthly temporal interval." to "...regridded to 1-degree spatial and monthly temporal resolution."

R: We changed it to "resampled" in page 9 line 3.

Lines 3-4: This sentence would be better written as "SSiB4/TRIFFID GPP was evaluated using the upscaled FLUXNET GPP (hereafter referred to as FLUXNET-MTE) (Jung et al., 2009; Jung et al., 2011)." Also where did you download the data from? Provide more information on how the dataset was created.

R: The sentence is re-written and additional information is provided (page 9 lines 4-10). The FLUXNET-MTE GPP was downloaded from https://www.bgc-jena.mpg.de/geodb/projects/Data.php. The FLUXNET observations of carbon dioxide flux were upscaled to the global scale using the machine learning technique, model tree ensembles (MTE), which was trained to predict site- level GPP based on remote sensing indices, climate and meteorological data, and information on land use.  This data set provides global monthly mean GPP at 0.5-degree spatial resolution for the period from 1982 to 2011. The FLUXNET-MTE GPP was regridded to 1-degree spatial resolution.

Line 7: Change "MTE-GPP data was remapped to 1-degree spatial and a monthly temporal resolution." to "FLUXNET-MTE GPP was regridded to 1-degree spatial and monthly temporal resolution.".

R: This sentience has been written in page 9 line 10.

Lines 9-10: Change this sentence to "In this study, SSiB4/TRIFFID was used to simulate the global vegetation distribution and assess the sensitivity of ecosystem trends to climate and eCO2."

R: Done in page 6 lines 5-6.

Line 10: Remove "For this purpose". "performed" instead of "conducted"

R: Done in page 6 line 6.

Lines 13-14: Remove the sentence "Meanwhile, the effect of largescale disturbance (LSD) on restricting tree expansion to savanna areas was investigated." Put this in the results/discussion sections.

R: Moved to the results/discussion sections.

Line 17: Remove "firstly". What are the multiple biotic variables?

R: "firstly" is removed. The multiple biotic variables stand for vegetation coverage, LAI and GPP. It has been replaced by "vegetation coverage, LAI and GPP" in the revised manuscript in page 6 line 14.

Pages 6-14

**3 Vegetation initial conditions**

This section should now be your results section. Each subsection should have the questions as the heading, so the reader does not have to go back to the introduction again. In each subsection, have a sentence that summarizes the main finding. This makes it easier for the reader to understand the key findings.

I suggest the following. Obviously, you should add your own question after each subsection number. I have only added what I think the section should be about.

3 Results

3.1 Evaluating the quasi-equilibrium simulation. Comparison to land cover map. Effect of large-scale disturbance.

3.2 Evaluating GPP/LAI.

3.3 Assessment of vegetation temporal variability.

3.4 External forcing effects on ecosystem trends.

R: The structure of results section has been modified accordingly in the revised manuscript.

3 Results

3.1 Vegetation initial condition

We added a sentence as the beginning of each subsection to point out the main content in this section.

5    The discussion section (section 4) should have 4 subsections, each discussing the corresponding results from section 3. The Conclusions section will be section 5.

R: The discussions currently in Section 3 have been moved to the discussion section.

**3.1 Quasi-equilibrium simulation**

10   Lines 3-16: The first 2 paragraphs could be moved to the discussion section.

R: Thanks for your suggestion. We have carefully read and compared before and after we move the first 2 paragraphs to the discussion section. We feel it is better to keep those paragraphs so the readers can easily follow this section, and the flow is better.

Lines 17-25: Move this paragraph to the beginning of this section.

15   R: Please see the answer for above question.

Line 11: Do you mean allocation instead of reallocation? Line 12: Don't start a sentence with "Figure X shows...". Make a statement regarding the result and reference the figure in brackets at the end of the sentence. Do this throughout the manuscript.

R: Yes, it is "allocation".  This has been corrected in the revised manuscript in page 10 line 1.

20   Thanks for pointing out, we have gone through manuscript to reduce use of this phrase.

Line 23: Delete the phrase "That being said".

R: Deleted.

Page 7 Lines 3-5: Delete the sentence "Detailed comparison of the simulated...".

R: Deleted.

25   Line 27: You could reference Figure 4 here.

R: We generated a figure includes vegetated area comparison in Supplement figure 1. It is cited in page 11 line 13.

Page 8 Line 28: Re-write this sentence as "The spatial correlation coefficient between model and FLUXNET-MTE GPP is 0.93 (P<0.05) (Figure 7)".

R: Modified as suggested in page 12 line 24.

Line 30: The standard way to quote global GPP values is in PgC/yr. Remove 1135 gC/m2/yr.

5 R: Modified in page 12 line 26 and page 13 lines 1-2.

Line 32: Lack of N-limitation.

R: Corrected in page 13 line 3.

Page 9 Line 8: Change "for the model validation" to "for model evaluation".

R: Changed in page 13 line 15.

10 Page 10 Line 6: I don't think you need Figure 9 as it is referenced only once. You could add a reference at the end of the sentence instead.

R: Done.

Lines 6-7: State the temperatures as Celsius instead.

R: The paragraph has been removed in the revised manuscript.

15 Lines 18-19: Remove the information regarding the stipples (dots) on Figure 10 as it is already included in the caption.

R: Done

Lines 24-25: Reference a figure at the end of this sentence.

R: This sentence refers to Figure 10 (Figure 7 in the revised manuscript). It is added in the revised

20 manuscript in page 14 line 17.

Page 11 Can you include a few sentences on how elevated CO2 affects GPP/LAI in the model as part of the discussion?

R: Yes. SSiB uses Collatz et al model to calculate photosynthesis process (Zhan et al, 2003) which is given the following equations:

$$A_n = \frac{g_b}{1.4} \frac{C_a - C_s}{p}$$

$$GPP = A_n + R_d$$

where $A_n$ is the net $CO_2$ assimilation, $g_b$ is stomatal conductance to latent and sensible heat transfer, $C_a$ is the atmospheric $CO_2$ concentration, $C_s$ is the $CO_2$ concentration at leaf surface, $p$ is the air pressure, and is $R_d$ the dark respiration rate of the canopy. Based on above equations, increase in $C_a$ leads to a larger $A_n$, then a larger GPP.

Vegetation in TRIFFID is presented as leaf, root, and wood carbon, and for each a corresponding carbon pool is updated the based on the net carbon available to it and the competition with other PFTs, which is controlled by Lotka-Volterra equations. LAI is calculated based on the leaf carbon pool and leaf phenology. Larger GPP leads more carbon goes in to the vegetation carbon pool in leaf, root and wood, then larger LAI.

A summary of these explanations has been added to the discussion section in page 20 lines 15-22.

Page 12 Lines 6-8: Remove these lines.

R: Removed.

**6. Conclusion**

I would like to see some more discussion in the Conclusions section. It is a bit short. Of the three sensitivity experiments, please state which one is the most important and why?

R: This is a valid point, we agree that the conclusion concerning driver contributions is a bit short. We have included additional discussion in the revised manuscript.

Line 9: Change "to the climate variability" to " to climate variability".

R: Changed in page 21 line 16.

Lines 10-11: Delete the phrase "The results show". Use "We have shown that the SSiB4/TRIFFID model can simulate the vegetation distribution and temporal variability for the X time period."

R: Done in page 21 lines 18-19.

Lines 28-30: Is there a University service for making the data available rather than a google drive account? Can you obtain a doi for the data? Also can you specify what data you have made available?

R: We have uploaded the available data to a University server and will indicate the available data in "data availability" section (page 21 lines 16-17). We will request a doi from your journal for the data when we submit the revision.

Figure 2 Can you use different colors for the lines as they are difficult to distinguish? Figure 4 I don't think you need this figure. Can you remove?

R: We have redrawn Figure 2 with different colors to make it clearer. Figure 4 has been moved to the appendix.

Figure 5 Can you include the MODIS dataset since you mention it on Page 4, line 24?

R: We have added figure to show the comparison of vegetated area in simulation, GLC and MODIS in supplement.

Figure 6 Please plot the differences (model - obs) instead.

R: This figure has been replaced.

Figure 7 Plot the difference instead.

R: We have redrawn this figure to show the difference.

Figure 8 You haven't said what the black line is? Is this SSiB4/TRIFFID?

R: Yes, it is SSiB4/TRIFFID. It has been added to the revised manuscript.

Change first line of caption to "Comparison of standardized LAI anomalies between simulation and observations for 9 sub-regions."

R: Done.

**Response to Reviewer 2**

**Main comments**

First, I fail to see what is new here compare to previously published studies: the current study uses only one model which does not seem to perform better than the TRENDY models used in Zhu et al. (2016) according to the results p. 8 l. 24-27 and p. 9 l. 22-25. As referee #1 mentions, this paper reads like a model evaluation and new scientific insights should be brought forward. If version 5 outperforms version 4 as mentioned p. 8 l. 25-27, the authors should consider using it instead.

R: There are a number of new features of our study. Firstly, it uses a new model, SSiB4/TRIFFID, which was not included in the TRENDY model inter-comparison. SSiB5/TRIFFID (as described on p. 8 l. 25-27) is a significantly updated version of SSiB4/TRIFFID. It is a new model which has not completed its testing. It will become publicly available next year. We have taken the sentence out in the text to avoid the confusion.

Secondly, and most importantly, our study focusses specifically on the impact of the climate regime shift during the 1980s. Previous DGVM studies (including TRENDY models used in Zhu et al, 2016), which investigated the effect of climate change on vegetation, only focused on the period after the 1980s. In our study, the contribution of the primary drivers on the ecosystem trends for each regime is identified. We estimate large-scale trends in terms of carbon fixation (GPP), vegetation growth (LAI), and expansion (vegetation fraction) rather than focus only on one aspect such as LAI trends in (Zhu et al, 2016).

Model inter-comparison exercises are excellent ways to assess common features of different model projections, and to estimate uncertainties. However, analysis of the responses to given climate anomalies is more effectively carried-out through detailed analysis of a single model, and this is precisely what we undertake in this paper. We have added text to the paper to make these innovative aspects of our study clearer.

Second, why is the study limited to the years 1958-2007? Considering the increasing availability of EO since 2007, extending the study period to nowadays would help address the "global vegetation variability" using satellite data as the title and the introduction (p. 3 l. 17).

R: We have downloaded three versions of Princeton meteorological dataset with the ending year of 2007 (v1x), 2010 (v1) and 2014 (v2.2), respectively. v1 had merged the data v1x plus the data from 2008-2010. However, when we compared the two versions (i.e. v1x and v1), we found that although v1x and v1 are generally consistent before 2007, however, there was an abrupt shift in some variables (such as wind speed) after 2007 (See Response Figure 2). To ensure the consistence and minimize the uncertainties could be involved by the meteorological forcing data, we decided to stop the simulation at 2007. The v2 data, which starts to be available in later 2016, is quite different from the v1 data (Response Figure 1, blue line) for a number of variables. Since by the time we have finished most of our work, we have kept with on the v1x data.

[Figure]

**Response Figure 2. Comparison between different version of Princeton meteorological datasets over global land (-180o W,180o E, -60o S, 75o N)**

Third, there is a lack of consistency between p-values reported, see for example p.9 l. 23 which points to possible cherry-picking from the authors.

R: Thanks for pointing out this. A consistent p-value of 0.05 is be used in the revised manuscript.

**Minor comments**

p. 2 l. 6 Can you support this statement with a reference?

R: References (Myneni et al., 1997; Piao et al., 2011, 2015; Ichii et al., 2013; Los 2013; Zhu et al., 2016) are added to the revised manuscript in page 3 lines 7-8.

p. 2 l. 9 Leaf area "per unit of ground" area

R: Corrected to "per unit ground area" according to Referee 1, Referee 2, and book "Plant Factory" Chapter 9: Photosynthesis and respiration (page 3 lines 10-11).

p. 2 l. 25 Consider citing Zhu et al. 2013 as an example of dataset covering the period 1980 to present

R: The citation was added in page 4 line 2.

p. 2 l. 30 Please cite articles that support this 'general consensus'

R: Ichii et al 2013; Piao et al 2015; Zhu et al 2016, 2017 have been added in the revised manuscript in page 4 lines 10-11.

p. 3 l. 15 See my main comment about the study period

R: Please see the response for the second main comment.

p. 3 l. 18 'apportioned' is perhaps more correct than 'attributed'

R: In this study, the experiments were designed to identify and quantify the contribution of three external forcings on the ecosystem trends. These drivers are the cause of these ecosystem trends. As a matter of course, previous studies on this subject also used the term "attribution", for instance Ichii et al 2013; Piao et al 2015; Zhu et al 2016, 2017. We feel the word "attribute" may be more familiar to readers.

p. 4 l. 19 Please define SPOT (Satellite Pour l'Observation de la Terre), indicate what type of sensor VEGETATION is and a what resolution these data were available.

R: We have added a definition of SPOT in the revised manuscript. The VEGETATION sensor has four spectral bands.. The spectral bands are blue (437–480 nm), red (615–700 nm), near-infrared (772–892 nm) and short-wave infrared (1600–1692 nm). This paragraph has been rewritten in the revised manuscript in page 7 lines 24-26 and page 8 lines 1-7.

p. 4 l. 26 To my knowledge GIMMS is also derived from AVHHR data.

R: Yes. The GIMMS LAI was generated using the overlapping AVHRR GIMMS NDVI3g data and best-quality MODIS LAI, then generating the full temporal coverage GIMMS LAI3g data using AVHRR GIMMS NDVI3g. The GIMMS LAI provides observation at 15-day temporal resolution and 1/12-degree spatial resolution for the period from July 1981 to December 2014. Proper modification was added in the revised manuscript in page 8 lines 16-28 and page 9 lines 1-3.

p. 5 l. 3-7 As the study aims to use satellite data, why not using MODIS GPP/NPP (or GIMMS-based NPP from Kolby-Smith et al. 2016).

R: This is a good suggestion. The observation-based datasets are used in this paper to evaluate the model simulation whenever the reference data available. The MODIS data set starts from 2000. The period is too short for this study. In fact, comparing the two observation-based GPP datasets (MODIS and FLUXNET MTE GPP), Anav et al. (2015) found they are in similar range of the global average and inter-annul variability, as well as in similar spatial pattern. FLUXNET-MTE is found to be more climate representativeness. Moreover, model simulations presented in Anav et al (2015) show higher spatial and temporal correlations against FLUXNET-MTE GPP than that against MODIS GPP.

p. 5 l. 20 onwards Can you summarize the experiments in a table?

R: A table is added in the revised manuscript.

p. 6 l. 8 Transient simulations are usually performed from a steady-state obtained under past conditions. Using the average conditions of the period 1948-2007 may reduce the model's sensitivity to the warming that occurred during that period. It would have been better to use the first ~10 years of driving data for this procedure.

R: Thank you, this is a good idea. We decided to use the average conditions of the period 1948-2007 in the quasi-equilibrium simulation was based on the practical consideration. Many experiments have been conducted to test the model performance under different meteorological condition. For instance, using the first year (1948) meteorological forcing, and the first 10/20 years' average. We obtain the best and the most stable results when the model was driven by average of the whole period 1948-2007, and also excluded the first 10-year results as spin-up period in the analysis.

p. 6 l. 15: Is this checked at pixel level, or only globally? Have you checked whether fluxes and initial stocks were at equilibrium? see e.g. Exbrayat et al. (2014) for the importance of initial stocks on transient simulations.

R: It is checked at reginal scale. We have checked the quasi-equilibrium status at different regions across the world, with particular attention on the transition areas between major climate zones. Per referee's concern, we draw the spatial distribution of absolutely relative changes of the last 10-year simulation for each PFT, which is defined as

$$average\left(\left|\frac{fraction - mean\_fraction}{mean\_fraction} * 100\%\right|\right),$$

where mean_fraction is the averaged from the last 10-year simulation, please see Response Figure 3.

As defined in the manuscript, the quasi-equilibrium status is reached when the fraction change is less than 2% of the mean vegetation fraction. Therefore, only fraction time series are shown here. We also checked other variables such as LAI and GPP at the equilibrium simulation, and both reach the quasi-equilibrium status. SSiB4/TRIFFID is a water, carbon and energy balanced model. When several key variables involved in water, carbon and energy cycle reach a steady-state systematically, other variables should follow. In the current version of SSiB4/TRIFFID, soil organic carbon mentioned in Exbrayat et al (2014) is a diagnostic variable as a result of plant litter and soil microbial process; no feedback to vegetation growth.

[Figure]

**Response Figure 3. Mean absolutely relative change of the last 10-year simulation in quasi-equilibrium simulation for each PFT**

p. 6 l. 22-25 Are these sentences referring to the model or GLC?

R: It was referring to the comparison between simulation and GLC. This sentence has been removed in the revised manuscript.

p. 7 l. 1 This part is very specific to the model used here. Readers who are not familiar with TRIFFID need a bit of context to understand how the LSD coefficient is used, and the impact of increasing its value ten-fold.

R: This is a valid point. LSD ($\gamma_v$) is a parameter in TRIFFID describing the rate of vegetation loss (units: $yr^{-1}$) caused by large scale disturbance such as fire, flooding and insect outbreaks. The change in plant functional type (PFT) fraction ($v$) is controlled by the PFT competition (the first term of the right-hand side of the equation) and the disturbance (the 2nd term).

$$C_v \frac{dv}{dt} = \lambda \Pi v \left\{ 1 - \sum_j c_{ij} v_j \right\} - \gamma_v v C_v,$$

where $v$ is the vegetation fraction for each PFT, $Cv$ is the carbon content in the plant, $\gamma_v$ is the large-scale disturbance which results in vegetated area loss at the prescribe rate. It was set to 0.004/$yr^{-1}$ for trees and 0.100/$yr^{-1}$ for grasses. Those values were chosen largely by model calibration in offline tests. We increased LSD in the tree and grass mixed areas as the consideration of more fire occurrences in those areas. We add some explanation in page 10 lines 15-25. We are developing the fire module to more realistically simulate that disturbance in the future.

p. 7 l. 21 You can also cite Poulter et al. (2014)

R: Done in page 19 line 22. Thanks.

p. 8 l. 19 Please clarify whether you are referring to global average LAI.

R: This sentence is regarding the LAI spatial distribution. The spatial correlation coefficients comparing observations are presented for both global and North Hemisphere.

p. 8 l. 25-27 This statement raises an important question: why do you use model version 4 when you know that model version 5 outperforms it?

R: As response to the main common 1, the SSiB5/TRIFFID mentioned in p. 8 l. 25-27 is an updated version of current SSiB4/TRIFFID. It is still under development. SSiB4/TRIFFID works well for the current study. We have removed this sentence in the revision.

p. 9 l. 22-23 Please consider rewording... correlations of 0.35 cannot be described as matching the reference data closely.

R: Thanks for pointing out. We have modified this sentence and only indicate it is a significant correlation in the revised manuscript in page 14 lines 1-2.

p. 10 l 19 Once again p-value...

R: Correction is made in the revised manuscript with consist p-value of 0.05.

5  p. 14 l. 29 I have not been able to access the data using this link, please check.

R: We have uploaded the data to a University server.

[revised manuscript text omitted]